# Understanding the Ability of LLMs to Handle Character-Level Perturbation

Anyuan Zhuo [1]  Xuefei Ning [2]  Ningyuan Li [2]  Jingyi Zhu [1]  Yu Wang [2]  Pinyan Lu [1]

## Abstract

This work investigates the resilience of contemporary large language models (LLMs) against frequent character-level perturbations. We examine three types of character-level perturbations including introducing numerous typos within words, shuffling the characters in each word, and inserting a large number of invisible characters into the text. Surprisingly, even under severe perturbation, such as shuffling nearly all words character-wise to produce text that is almost unreadable to humans, or inserting invisible characters which are several times more than the visible ones as noise, many LLMs still maintain notable performance. We explore the underlying causes of this robustness and find that LLMs exhibit remarkable resilience to chaotic segmentation and fragmented tokenization. Furthermore, we examine the mechanisms by which LLMs remove perturbations to correctly comprehend text, including both implicit and explicit mechanisms for character-level perturbation. We hope that our findings on the low-level robustness of LLMs will unveil their inherent architectural strengths, reveal the potential risks of their misuse, and inform the reliable deployment of LLMs across diverse application scenarios.

## 1. Introduction

Large language models (LLMs) exhibit remarkable performance across a wide range of tasks, motivating substantial research efforts to study their behavior and mechanisms. A widely adopted approach for both evaluating model behavior and probing underlying mechanisms is to introduce controlled perturbations to the input and examine how the model outputs change accordingly. By varying various attributes of the input query, such as semantic descriptions (Xie et al.,

2025), task-relevant information (Srivastava et al., 2024; Yuan et al., 2024), problem structure (Ye et al., 2025; Xie et al., 2025), or other contextual conditions (Jiang et al., 2024), one can assess the ***high-level robustness*** of LLMs against these high-level perturbations and use the observed behavioral changes to test hypotheses about what internal mechanisms the model may rely on.

Existing perturbation-based mechanism studies primarily focus on high-level perturbations. While several works (Rocamora et al., 2024; Yi et al., 2024; Zhu et al., 2023; Tao et al., 2024) leverage character-level perturbations to test the ***low-level robustness*** of LLMs, their focus lies in the design of adversarial attack methods corresponding to certain application scenarios, with each work proposing some perturbation strategies and reporting performance. As a result, there is no comprehensive understanding of how various types of character-level perturbations affect *model performance*, and their impacts have neither been studied at the level of *fine-grained model behavior* nor examined through *internal diagnosis*. This gap motivates our work, which studies the following research question:

> *How well do contemporary LLMs handle different types of character-level perturbations, and how do they handle them?*

We investigate three types of character-level perturbation methods with varying perturbation strengths: (i) **Typo-Method** (Li et al., 2018): "typo"-like perturbations, including deletion, substitution, swapping, and insertion of a single character within selected words, which commonly arise in human-typed or OCR-produced texts; (ii) **Shuffle-Method**: a synthetic perturbation that randomly shuffles characters within selected words, introducing severe disruption of character order that is very hard for human to process; (iii) **UCC-Inj**: a perturbation that injects invisible Unicode control characters between adjacent visible characters, introducing substantial non-semantic noisy characters and causing severe tokenization fragmentation. These perturbation methods are summarized in Table 1.

All three perturbations alter the tokenization outcome. In particular, Shuffle-Method and UCC-Inj can completely corrupt the tokenization, producing a character-level tokenized sequence. Moreover, UCC-Inj introduces a large amount

[1]Shanghai University of Finance and Economics, China [2]Tsinghua University, China. Correspondence to: Xuefei Ning <foxdoraame@gmail.com>, Pinyan Lu <lu.pinyan@mail.shufe.edu.cn>.

*Proceedings of the 43rd International Conference on Machine Learning*, Seoul, South Korea. PMLR 306, 2026. Copyright 2026 by the author(s).

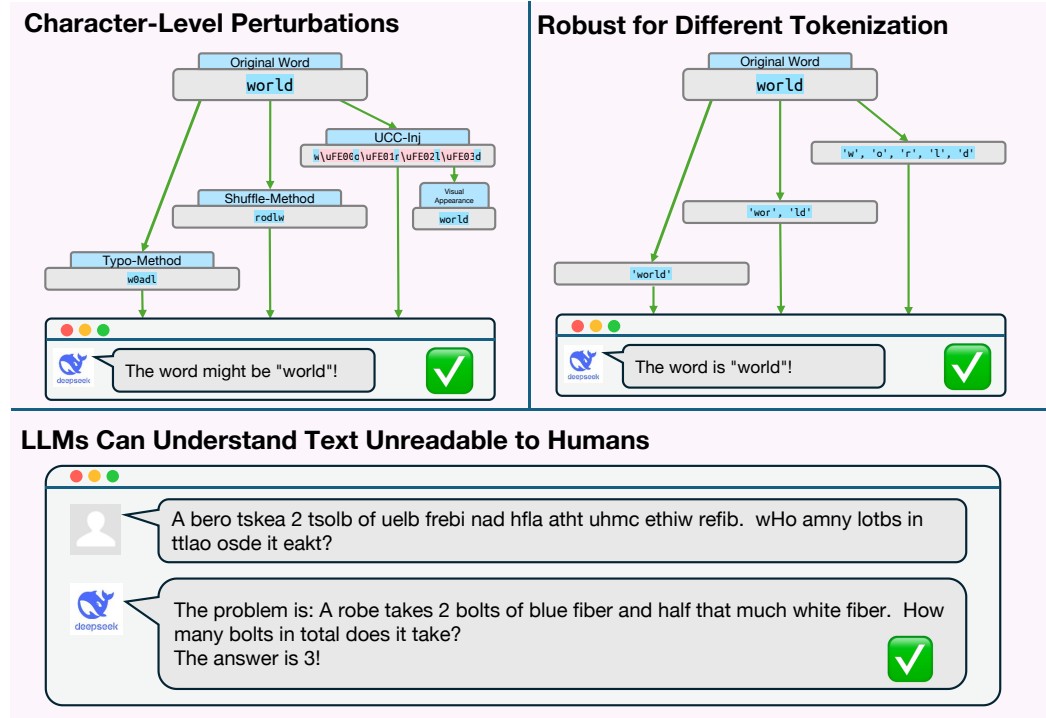

*Figure 1.* Examples of character-level perturbations and models' robustness to abnormal tokenization and perturbed text

*Table 1.* Summary of the character-level perturbation methods studied in this work. Despite differing practical motivations, all perturbations are used as controlled probes to analyze LLM robustness under tokenization corruption and character-level noise.

| Perturbation Type | Method Description | Strength Control | Tokenization Corruption | Noisy Character Introduction | Human Readability | Application Scenario |
|---|---|---|---|---|---|---|
| Typo-Method | Character-level deletion, substitution, swapping, or insertion applied to selected words | Ratio of perturbed words | Medium | Medium | Medium | Naturally occurring human typos, OCR errors |
| Shuffle-Method | Random shuffling of characters within selected words | Ratio of shuffled words | Strong | None | Low | Synthetic stress test; limited direct practical use |
| UCC-Inj | Injection of invisible Unicode control characters between adjacent visible characters | Number of injected control characters per visible character | Strong | Strong | High (no effect) | Adversarial text obfuscation, detection evasion, and security-related scenarios |

of non-informative characters. Perturbation strength is controlled by the proportion of affected words for Typo-Method and Shuffle-Method, and by the number of injected invisible characters per visible character for UCC-Inj.

In this work, we identify a seemingly counter-intuitive observation: *LLMs retain a non-trivial level of comprehension even when the perturbed text becomes entirely unreadable to humans under Shuffle-Method, or when per-character injection of invisible noise in UCC-Inj fully fragments tokenization while introducing substantial non-informative characters.* An illustrative example is shown in Figure 1. This phenomenon highlights a previously underexplored aspect of LLM behavior—their ability to tolerate severe tokenization corruption and high levels of character-level non-informative noise. We then analyze in detail *how* LLMs

behave under such perturbations and *what factors and mechanisms may underlie this seemingly surprising robustness.*

The remainder of this paper is organized as follows. In Section 2, we elaborate on the character-level perturbation methods. Section 3 then presents model performance under different perturbation types and strengths. For a fine-grained behavioral analysis, Section 4 examines how model behavior varies with respect to three factors: (i) *model*, (ii) *problem*, and (iii) *noise*. Next, Section 5 investigates the mechanisms underlying robustness by separately evaluating the model's abilities in **handling abnormal tokenization** and **denoising character-level noise**, and by testing denoising-related hypotheses to shed light on how LLMs cope with such perturbations. Finally, findings are summarized in Section 7.

## 2. Perturbation Methods

We introduce three types of character-level perturbations as described below.

**Typo-Method** simulates common typing errors by applying four operations sequentially to each selected word:

- **Deletion**: Remove one internal character, preserving the first and last.
- **Substitution**: Replace one character with a visually or spatially similar one (see Table 17).
- **Swapping**: Exchange two adjacent characters.
- **Insertion**: Insert a random English character.

We apply the four typo error methods above to perturbed words in order. Each method is applied to each word exactly once.

**Shuffle-Method** randomly permutes all characters within perturbed words, making them nearly unreadable.

We only apply these methods on the pure-words which contain only English letters with length greater than 2. We apply the method on 30, 50, 70, and 100 percents of the words that meet these restrictions, donoted by *30-Typo*, *50-Typo*, *70-Typo*, *100-Typo* for Typo-Method, and *30-Shuffle*, *50-Shuffle*, *70-Shuffle*, *100-Shuffle* for Shuffle-Method.

**Unicode Control Character Injection (UCC-Inj)** inserts invisible Unicode Variation Selectors (VS) after each visible character (including spaces). We insert 1, 2, or 3 VS characters per visible character, denoted as *1-Inj*, *2-Inj*, *3-Inj*.

UCC-Inj disrupts models via: (1) tokenization fragmentation, forcing words to be split into individual characters interleaved with VS tokens (e.g., "the" becomes [t, ?, h, ?, e, ?]), and (2) noise injection, where VSs obscure the input.

## 3. Result

We evaluate our attack methods on **GSM8K** (Cobbe et al., 2021), which is a problemset of grade school math word problems. For each problem, we apply the perturbation methods mentioned in Section 2 to the problem statement, and prepend a standard user prompt (see Table 15, not perturbed, to stimulate a user directly input direction), as the input to models. We test DeepSeek (DeepSeek-AI, 2024) and Qwen3 (Bai et al., 2023) and Llama (Grattafiori et al., 2024); results are in Figure 2 (detailed scores in Section B.1). As a result, **The attack methods do reduce model accuracy. . but even some under strong perturbation, such as *100-Shuffle* and *3-Inj*, some models, such as DeepSeek-Reasoner, still retain a partial understanding to the input statement, which shows a notable robustness under such character-level perturbations**. Moreover, for weaker perturbations, such as *30-Typo*, *30-Shuffle* and *1-Inj*, most of the models retain a high accuracy rate. For example,

*Table 2.* SQuAD score of Qwen3-30B (A3B) under nothinking mode

| Perturbation | Strength | score-em[1] | score-f1[1] | score-em[2] | score-f1[2] |
|---|---|---|---|---|---|
| Baseline | 0 | 63.2 | 81.48 | 63.87 | 83.14 |
| Typo-Method | *30-Typo* | 30.67 | 51.18 | 29.6 | 54.38 |
| | *50-Typo* | 17.27 | 33.31 | 16.27 | 36.79 |
| | *70-Typo* | 5.6 | 13.25 | 6.2 | 17.71 |
| | *100-Typo* | 0.33 | 1.39 | 0.47 | 2.09 |
| Shuffle-Method | *50-Shuffle* | 35.2 | 56.26 | 34.47 | 59.49 |
| | *70-Shuffle* | 24.87 | 42.84 | 24.27 | 47.3 |
| | *100-Shuffle* | 14.0 | 27.16 | 16.53 | 47.3 |
| | *30-Shuffle* | 6.07 | 11.18 | 7.93 | 16.05 |
| UCC-Inj | *1-Inj* | 7.67 | 9.23 | 20.47 | 34.0 |
| | *2-Inj* | 0.73 | 1.02 | 1.33 | 2.37 |
| | *3-Inj* | 0.2 | 0.39 | 0.53 | 0.74 |

[1] no-thinking mode
[2] thinking mode

Qwen3-30B(A3B) gets the accuracy of 61.6% under *30-Typo*, 81.5% under *30-Shuffle* and 57.5% under *3-Inj*.

Beyond mathematical tasks, we also evaluate the performance of Qwen3-30B(A3B) under character-level perturbations on SQuAD (Rajpurkar et al., 2016), a reading comprehension dataset. We uniformly sample 1,500 instances from the validation set, with the results illustrated in Table 2. Similar to the observations on GSM8K, the model's performance on this dataset declines as the perturbation intensity increases, yet it still maintains a reasonable level of comprehension. Meanwhile, the model continues to exhibit stronger robustness against the shuffle method.

## 4. Factors Influencing the Performance of LLMs

We investigate the factors that influence LLM performance under our character-level perturbations, including (i) *Model-related*, (ii) *Problem-related*, and (iii) *Noise-related*.

### 4.1. Model-Related Factors: Model Family and Model Size

As shown in Figure 2, we find that **different model families have vastly different performances under attack**. Generally, Qwen3 and DeepSeek exhibit a greater robustness, while Llama family models have a poorer performance than other models. For example, Llama3.1-70B get accuracy of 10.0% under *1-Inj*, which is significantly lower than 43.4%, the accuracy that a smaller model Qwen3-30B(A3B) get.

We also evaluate the performance of different size within the model family Qwen3, and we find that **larger models consistently exhibit greater robustness than smaller ones**, as shown in Figure 3. For instance, Qwen3-32B got accuracy of 53.6% under *1-Inj*, which is higher than 43.4%, the result of Qwen3-30B Model.

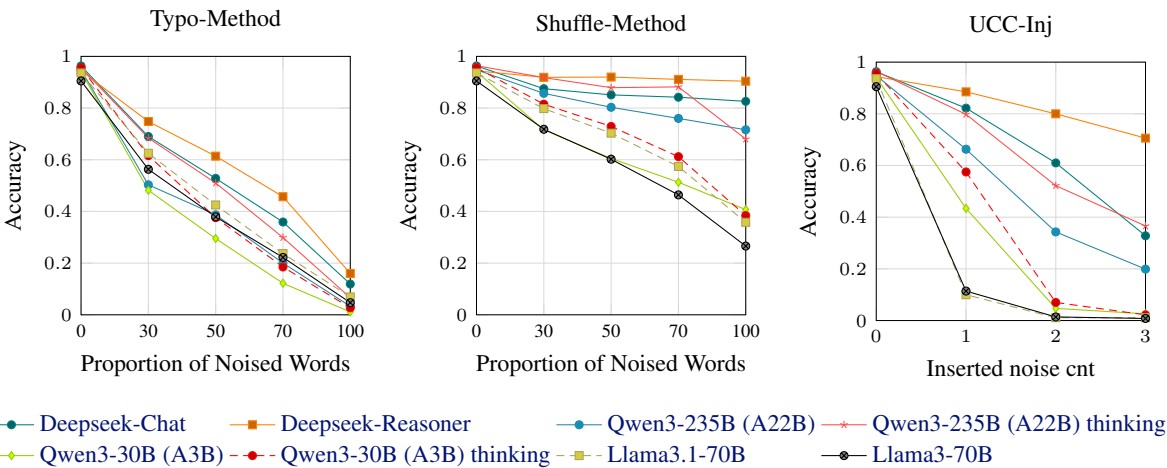

*Figure 2.* The results of 3 attack method on GSM8K datasets. In the legend, "Qwen*" without "thinking" indicates non-thinking mode.

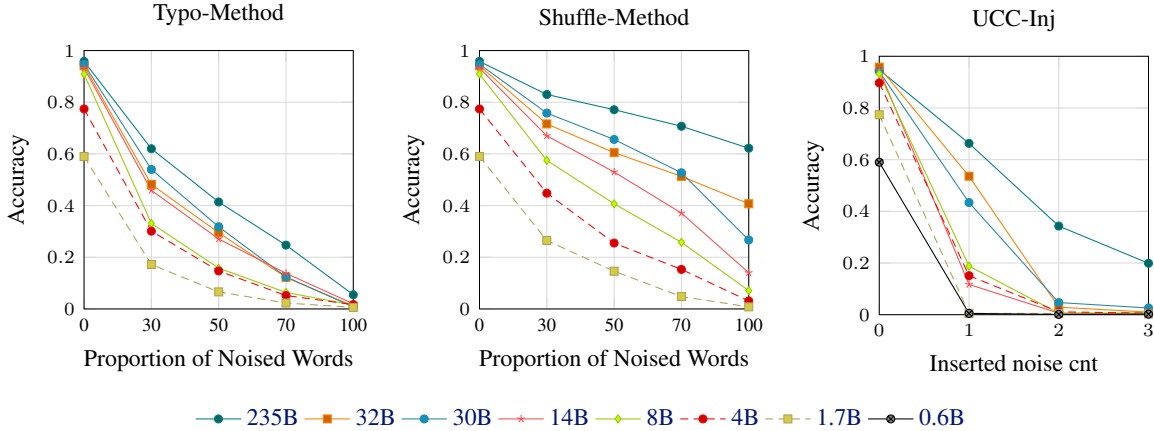

*Figure 3.* Result of different size Qwen3 models under perturbations, all results are in nothinking mode.

## 4.2. Problem-Related Factor: Statement Length

In this section, we investigate the models' performance when input statements become longer. In order to keep the difficulty of problems unchanged while increasing the statement length, we construct three prefixes of increasing length, labeled prefix1, 2, and 3. Each of them only provides contextual background for problems but contain no information useful for solving them (details in Appendix F.1).

We test these augmented problems with Qwen3-30B (A3B)(nothinking) and show the results in Table 3. **The models are robust to contextual prefixes alone as expected, and are also robust to Typo-Method and Shuffle-Method**. The performance is kept or even better when the statement becomes longer. **But once UCC-Inj is applied, longer noised inputs lead to greater performance degradation**.

*Table 3.* GSM8K accuracy of Qwen3-30B (A3B) under nothinking mode with different problem prefixes.

| Perturbation | Strength | no prefix | prefix1 | prefix2 | prefix3 |
|---|---|---|---|---|---|
| Baseline | 0 | 0.941 | 0.936 | 0.935 | 0.932 |
| Typo-Method | *30-Typo* | 0.482 | 0.582 | 0.614 | 0.591 |
| | *50-Typo* | 0.295 | 0.406 | 0.388 | 0.370 |
| | *70-Typo* | 0.122 | 0.229 | 0.211 | 0.224 |
| | *100-Typo* | 0.011 | 0.042 | 0.033 | 0.020 |
| Shuffle-Method | *30-Shuffle* | 0.716 | 0.773 | 0.766 | 0.776 |
| | *50-Shuffle* | 0.605 | 0.672 | 0.683 | 0.664 |
| | *70-Shuffle* | 0.513 | 0.569 | 0.558 | 0.583 |
| | *100-Shuffle* | 0.407 | 0.014 | 0.008 | 0.01 |
| UCC-Inj | *1-Inj* | 0.434 | 0.370 | 0.289 | 0.052 |
| | *2-Inj* | 0.047 | 0.041 | 0.021 | 0.006 |
| | *3-Inj* | 0.026 | 0.020 | 0.017 | 0.003 |

## 4.3. Noise-Related Factors

### 4.3.1. INDIVIDUALLY APPLYING EACH METHOD USED IN TYPO-METHOD

For the Typo-Method, we applied the four perturbation methods mentioned in Section 2 to the statement in sequence. In order to investigate the inference ability of each method, we apply each method individually to the statement. As shown in Figure 4, **each individual typo method is capable of perturbing models. Among these methods, deletion exhibits the strongest perturbation effect while insertion exhibits the weakest. And none of the methods is as disruptive as Typo-Method**. Each of the methods decreases model's performance no more than 40%, but when used together, they almost entirely prevent the models from understanding the statement.

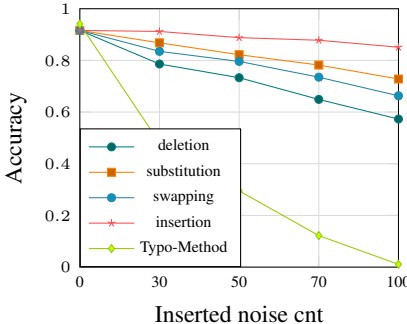

*Figure 4.* The effects of individually applying each of the four methods used in the typo method separately.

### 4.3.2. VS SET SIZE FOR UCC-INJ

There are 256 VS characters. In UCC-Inj, we randomly sample VS characters from the full VS set, which demonstrates effectiveness. We further evaluate a variant that randomly samples VS characters from a smaller subset. We show the results with Qwen3-30B (A3B) in Figure 5. **A larger VS set size leads to stronger attack effects**. For example, model accuracy is 67.1% when the set size is 1, but drops sharply to 2.2% when the full set of 256 is used. This indicates that as the noise variety increases, it becomes harder for models to recognize the noise pattern and perform denoising.

### 4.3.3. REPLACING VS WITH 'A' IN UCC-INJ

We replace VS characters with 'A', a visible noise characters, and show the results in Figure 6. Overall, **visible noise character 'A' cause greater performance degradation than invisible ones**. Inserting 'A' (equivalent set size=1) leads to accuracy drops comparable to VS with set size = 1 under *1-Inj* and *2-Inj*. But an interesting anomaly arises under *1-Inj* when VS is replaced with 'A': model accuracy drops to just 3.2%, even much lower than under *2-Inj*. We attribute this to tokenization errors, i.e., we find that the

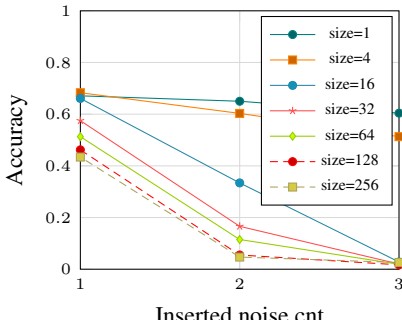

*Figure 5.* GSM8K accuracy of Qwen3-30B (A3B)(no-thinking) with different VS set size.

inserted 'A' is often merged with adjacent characters into single tokens by the tokenizer, which disrupts interpretation more than the character-wise tokenization fragmentation caused by inserting VS. To validate this explanation, we enforce character-level tokenization to the text applied with 'A' noises. As shown in Figure 7, the anomaly disappears and the performance aligns with expectations.

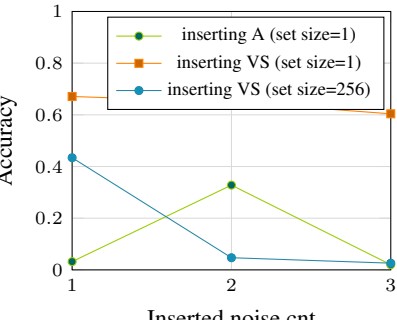

*Figure 6.* Replace VSs with 'A'

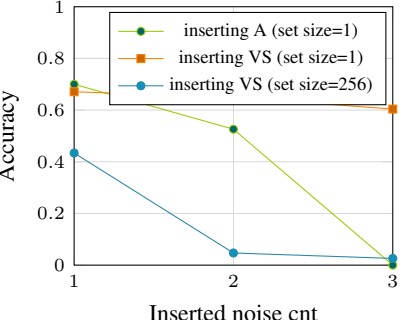

*Figure 7.* Applying 'A' as noise with character-level tokenization

# 5. Mechanism of Handling Character-level Perturbations

This section investigates the mechanisms through which LLMs comprehend inputs perturbed by noise characters. We break down the ability of handling the perturbed input into two aspects: (i) the robustness to **abnormal tokenization**; (ii) the ability in **denoising character-level noises**.

## 5.1. Robustness to Abnormal Tokenization

### 5.1.1. ABILITY IN HANDLING CHARACTER-LEVEL TOKENIZATION

All of our methods exert a substantial influence on models' tokenization process. For perturbed words, which are often out of vocabulary, the models no longer see a complete word (even an erroneous one) but rather a sequence of fragmented, nearly character-level tokens. Regardless of the specific process or mechanism the models use to remove perturbations, they must, at some stage, recover the word-level meaning directly from these fragmented tokens to answer correctly. That is, the models must learn the mapping from several single-character tokens to their corresponding complete word.

To examine whether LLMs can still comprehend text under character-wise tokenization, we convert each GSM8K problem into a token sequence where every character is treated as a separate token. These sequences are then fed into Qwen3-30B and Llama3.1 models, with results reported in Table 4 (left column).

*Table 4.* GSM8K accuracy between abnormal and normal tokenization.

| Model | Character Level | Random Segmentation | Normal |
|---|---|---|---|
| Qwen3-30B (A3B) | 0.914 | 0.935 | 0.941 |
| LLama3.1-70B | 0.883 | 0.932 | 0.936 |

We observe that: (i) **LLMs retain substantial comprehension ability under character-wise tokenization**. Models show only a small or moderate degradation of less than 10%. (ii) Larger models are more robust to character-wise tokenization (results for smaller of models are shown in Table 12. (iii) Qwen3 models outperform Llama3 models in handling character-wise tokenized inputs. For instance, Qwen3-30B(A3B) achieves 91.4% accuracy under character-wise tokenization (94.1% under normal tokenization), whereas the much larger Llama3.1-70B drops from 93.6% to 88.3%. (iv) Inspection into the LLMs' outputs reveals that LLMs neither explicitly assert the anomalous tokenization nor attempt to reconstruct a normal token sequence during CoT.

### 5.1.2. ABILITY IN HANDLING INCORRECTLY SEGMENTED TOKENS

While Section 5.1.1 demonstrates LLMs' robustness to character-level fragmentation, methods such as Shuffle-Method often produce a more complex form of tokenization noise. Specifically, shuffling characters within a word does not guarantee that the model will tokenize the result into individual characters; instead, it frequently yields several multi-character fragments (e.g., "apepl" may be tokenized as ["ap", "e", "pl"]). This observation motivates a deeper investigation: beyond pure character-level breakdown, can LLMs comprehend text when words are incorrectly segmented into arbitrary, non-character subword units?

To probe this, we design a tokenization scheme that explicitly forces incorrect mid-word splits. For each word, a random split point (excluding the first and last character) is chosen, dividing the word into two substrings. Each substring is then independently processed by the model's standard tokenizer, and the resulting token sequences are concatenated. For instance, "apple" might be split into "app" and "le", and might be tokenized further as ["app", "l", "e"].

This setup introduces a two-hop comprehension challenge that differs fundamentally from character-level tokenization. Character-wise tokenization requires a direct mapping from a sequence of single-character tokens to the intended word, while randomized segmentation demands that the model first infer the character composition of each incorrectly formed token, and then mentally reassemble these tokens to recover the original word. This process tests the model's capacity for morphological reasoning over corrupted token boundaries.

We evaluate Qwen3-30B(A3B) and Llama3.1-70B on GSM8K under this randomized segmented tokenization. The results are presented in Table 4 (middle column).

The result shows that **LLMs exhibit notable, though reduced, robustness to random in-word segmentation**. For example, Qwen3-30B(A3B) keeps a high accuracy of 93.5% under randomized segmentation tokenization, compared to 94.1% under normal tokenization on clean text.

Based on the above experiments, we find that **models exhibit strong robustness to abnormal tokenization on clean text**. Performance degradation under these conditions is minimal; for instance, Qwen3-30B(A3B) maintains over 90% accuracy with both forms of abnormal tokenization. However, as evidenced by the results in Figure 6 and Figure 7, model performance drops precipitously when adversarial noise is introduced, specifically, when noise characters (e.g., the inserted A) are frequently merged with original characters into the same token during tokenization.

## 5.2. Ability in Handling Character-Level Perturbation

To understand how LLMs understand the character-level noised input, we propose a hypothesis by analogy to human solving process: *The LLM first rewrites a clean problem in the CoT process explicitly by a denoising process that identifies the noised pattern, and then iteratively skip the noise, copy informative characters, and then answers the clean problem*. Testing this hypothesis involves answering the following three questions:

1. **Explicit problem rewriting or not**: Does the LLM attempt to explicitly rewrite the clean problem in the output text during its CoT process? Is the rewriting correct?
2. **Relationship between explicit rewriting and successful solving**: Does explicitly rewriting the clean problem positively correlate with the odds of the LLM solving the problem? If so, does guiding the LLM to explicitly rewrite the problem help?
3. **Rule-based or implicit procedure for rewriting**: Our perturbations are rule-based. Does the LLM recognize the underlying rules of perturbation and execute a rule-based procedure to explicitly understanding the statement?

### 5.2.1. EXPLICIT PROBLEM REWRITING OR NOT

To investigate the first question, we calculate the proportion of responses containing explicit rewritting across all problems. Specifically, the model's CoT outputs are categorized into three types:

A. The model correctly produces clean (noise-free) text during its response (preserving the original meaning), followed by a reasoning process (regardless of correctness).
B. The model produces clean text, but the denoised output contains errors or omits critical information, followed by a reasoning process and an answer (regardless of correctness).
C. The model does not explicitly output clean text and directly provides a reasoning process (or only an answer without explicit reasoning).

We analyze GSM8K responses under UCC-Inj from Qwen3-30B (A3B) in the no-thinking mode, using Qwen3-30B (A3B) to assist categorization under the above criteria (see prompts in Section C.3). As shown in Table 10, for *1-Inj*, 67.3% of Qwen3's responses fall into category A or B. This suggests that **LLMs do rewrite problems under UCC-Inj in the CoT**. Furthermore, when explicitly instructed to remove noisy characters (see prompts in Table 16) or provided with a 1-shot in-context learning (ICL) example of explicit rewriting, the proportion of responses involving explicit rewriting (A or B) increases substantially – to 99.5%

(w. instruction) and 99.9% (1-shot). Models' response type under *2-Inj* and *3-Inj* are shown in Figure 8.

For Typo-Method and Shuffle-Method, we also analyze Qwen3-30B (A3B)'s responses and find that for these methods except *70-Typo* and *100-Typo*, **almost every noised problem is rewritten in CoT**, as shown in Table 11. The observed difference in the model's rewriting strategy stems from the perceptibility of the perturbations. For visible perturbations, the noise is readily apparent, making the explicit rewriting of a clean problem a natural and logical step within the problem-solving narrative. In contrast, invisible perturbations leave the visual rendering intact for a human reader. Consequently, an explicit denoising step in the CoT would appear cognitively incongruous, as the input text ostensibly contains no obvious errors to correct, leading to a lower tendency for the model to perform explicit rewriting under such invisible attacks. For *70 and 100-Typo*, the perturbation is so strong that models can't reconstruct the underlying word, thus models can't successfully rewrite the statement. Overall, the results suggest that **LLMs do rewrite problems in the CoT under all three kinds of perturbations**.

### 5.2.2. RELATIONSHIP BETWEEN EXPLICIT REWRITING AND SUCCESSFUL SOLVING

Regarding the second question, the first row of Table 10 shows that for UCC-Inj, **attempting explicit rewriting is positively correlated with the odds of solving a problem successfully**: the accuracy of responses in categories A or B is $\frac{20.9+12.5}{22.4+44.9} = 49.6\%$, compared to only 33% for category C. Tables 5 and 10 also show that **when models are explicitly instructed to remove noisy characters or provided with a 1-shot example, the proportion of type A and B responses increases, and overall accuracy improves**. For example, under *1-Inj*, the accuracy of the thinking model rises from 57.5% to 72.8% (w. instruction) and 78.8% (1-shot). See Table 14 for the results of LLama-3.1-70B.

*Table 5*. GSM8K accuracy of Qwen3-30B (A3B) with different prompt strategies.

| Prompt Strategy | Baseline | *1-Inj* | *2-Inj* | *3-Inj* |
|---|---|---|---|---|
| w.o. instruction | | 0.434 | 0.047 | 0.026 |
| w. instruction | 0.941 | 0.660 | 0.095 | 0.031 |
| w. 1-shot | | **0.788** | **0.208** | **0.085** |

A closer look reveals that 158 problems ($\approx 12\%$) that yield incorrect answers in category C without instruction shift to A with correct answers under explicit instruction, and 196 problems ($\approx 15\%$) make the transition with 1-shot example.

For Typo-Method and Shuffle-Method, since the model rewrites the text in most responses, to verify the impact of rewriting on accuracy improvement, in addition to ex-

plicitly instructing model to rewrite the problem, we also test the result when explicitly instructing model NOT to rewrite the problem. The result are shown in Table 6. For *30,50,70-Shuffle* and *30-Typo*, we find that **rewriting the statement in the CoT does improve models' accuracy**. But **for stronger perturbation**, that models get a lower accuracy on baseline, **rewriting does not further improve model correctness**, as the text becomes too garbled for the model to reconstruct the true meaning through rewriting.

*Table 6.* GSM8K accuracy of Qwen3-30B (A3B) with different prompt under Typo-Method and Shuffle-Method

| Method | baseline | Must-Rewrite | No-Rewrite |
|---|---|---|---|
| *30-Typo* | 0.482 | **0.487** | 0.476 |
| *50-Typo* | **0.295** | 0.253 | 0.281 |
| *70-Typo* | 0.122 | 0.091 | **0.145** |
| *100-Typo* | 0.011 | 0.012 | **0.029** |
| *30-Shuffle* | 0.716 | **0.760** | 0.688 |
| *50-Shuffle* | 0.605 | **0.646** | 0.566 |
| *70-Shuffle* | 0.513 | **0.518** | 0.431 |
| *100-Shuffle* | **0.407** | 0.345 | 0.350 |

We find that the rewritten problem in the CoT is generated word- or subword-wise rather than character-wise. For example, given a *1-Inj* perturbed sequence [t, ?, h, ?, e, ?], the model produces 'the' as a single token instead of outputting [t, h, e]. This behavior persists even when we apply character-wise tokenization to the response in the 1-shot example. Therefore, we can conclude and answer the third research question: **Rewriting clean text in the CoT does not assist the model in handling perturbations at the word level**. Instead, it serves primarily to reconstruct the meaning of the problem at high-level (i.e., the semantic level), thereby enhancing the model's overall comprehension of the question. **At the low-level (i.e., the word level), the model directly processes the perturbed tokens through an internal, implicit mechanism to strip away noise and recover the meaning of words**, but not a rule-based mechanism.

### 5.2.3. IMPLICIT PROCEDURE FOR HANDLING PERTURBATION AT LOW LEVEL

Based on the discussion so far, the results are contrary to our initial hypothesis, which suggests that the model comprehends the semantic meaning of the noise-perturbed text before reconstructing the clean problem. We therefore hypothesize that the model employs an implicit denoising mechanism to extract the underlying meaning of each word as the input is processed layer by layer. The implicit dealing with different perturbation is different.

To verify the existence of such a layer-by-layer denoising mechanism, we construct a dataset comprising both positive and negative samples, where each instance is an English

sentence of approximately 20 words. Positive examples terminate with a specific target word (e.g., "river"), while negative examples end with a related but distinct word (e.g., "lake" or "bay"). We allocate 80% of the data for training and the remaining 20% for testing. For each layer of Qwen3-30B(A3B), we train a linear probe. The input to the probe is the hidden state of a given layer during the forward pass, and the objective is to determine whether the sentence ends with the target word. These layer-wise classifiers are trained exclusively on the clean training set.

We then apply character-level perturbations to the test set and feed the resulting hidden states from the forward pass into the corresponding probes. According to our hypothesis, **classification accuracy should be low in the shallow layers. As the layer depth increases, the model's internal mechanisms should complete the denoising process and capture the underlying semantics, leading to an observable increase in accuracy**. Our experimental results align well with this hypothesis. Figure 10 presents a case study using "river" as the target word.

While formalizing and validating the hypothesis is challenging for a computational system with tens of billions of parameters and highly nonlinear dynamics, we make some initial attempts as follows.

**For UCC-Inj**, the model appears to learn to focus on non-noise characters to recover semantics. Attention visualization (Figure 9) reveals **specific heads that predominantly attend to clean characters**. As shown in Table 3, longer noisy inputs degrade performance. We further track the evolution of feature similarity between clean and noisy inputs: for each token prefix ending in a complete word, we compute the average cosine similarity of hidden states (excluding the final layer) across layers. Similarity declines as the prefix lengthens, indicating that **the denoising mechanism is implicit and not fully robust**, rather than a stable rule-based procedure that would be invariant to additional noise tokens from the same pattern.

**For Shuffle-Method**, models show notable robustness to severe within-word character shuffling. Although this perturbation is visually confusing to humans, it neither introduces additional noise characters like UCC-Inj nor modifies the original information like Typo-Method, which could lead to information loss or the introduction of errors. Aided by its robustness to abnormal tokenization, the model can implicitly reassemble the original word without explicit sequential reconstruction, maintaining comprehension despite local order variance.

**For Typo-Method**, performance drops sharply under high perturbation (e.g., *100-Typo*), because typo operations (substitution, deletion, insertion) directly alter or remove characters, erasing part of the input's information. Unlike shuf-

fling—which preserves the character set—typos modify the inventory itself, obscuring word identity. Compared to the fixed, learnable noise pattern in UCC-Inj, typo perturbations are irregular and diverse, providing no consistent structural cue. In settings below *100-Typo*, the model might partly recover meaning from contextual clues in unperturbed words, but as the perturbation ratio rises, contextual recovery weakens, exposing the limits of implicit low-level reconstruction from incomplete or altered character sets.

## 6. Related Work

**Injecting VS Characters Techniques.** Butler (2025) proposed inserting VS characters into LLM inputs to encode arbitrary information, noting that LLMs can process these invisible characters while the text remains normal to human readers. If informed of the encoding scheme, models can partially decode the embedded information.

**Adversarial Attacks on LLMs.** The attacks craft inputs to cause LLMs to produce incorrect or harmful outputs. Branch et al. (2022) and Perez & Ribeiro (2022) demonstrated that even simple manual prompt injections can divert models from their original tasks, Rajeev et al. (2025) showed that adding irrelevant textual triggers to math problems can induce incorrect answers. Zhu et al. (2023) and Tao et al. (2024) researched LLMs' robustness under different level adversarial attacks, including character-level perturbations by introducing typos or errors to words, and found that models keep partial robustness under the attack.

**LLM's Robustness under Abnormal Tokenizations.** Zheng et al. (2026) researched the models' robustness under abnormal tokenizations, and pointed that they keep a high performance under character level and randomly sampled tokenization. This work found that models grasp the semantics of non-canonical tokenizations, and this robustness arises in the instruction-tuning phase.

## 7. Conclusion

This work systematically investigates the resilience of contemporary LLMs against extensive and structured character-level perturbations. We introduced and evaluated three primary attack methods, Typo-Method, Shuffle-Method, and UCC-Inj, designed to disrupt lexical integrity, tokenization, and human readability to varying degrees. Surprisingly, even under severe perturbations that render text nearly incomprehensible to humans, such as character-wise shuffling, or inserting multiple times more invisible noise than visible content, many LLMs retain notable performance on reasoning tasks like GSM8K.

Our key findings are as follows:

1. **Robustness to Abnormal Tokenization**: LLMs demonstrate significant robustness to abnormal tokenization, including pure character-level tokenization and randomized in-word segmentation.
2. **Dual-Layer Resilience Mechanism**: the ability to handle character-level perturbation involves two level mechanisms: (i) a **high-level, explicit ability** to rewrite a clean version of the perturbed problem within CoT process, which significantly correlates with and improves final task accuracy when guided; and (ii) a **low-level, implicit ability** to directly extract word-level meaning from corrupted token sequences through the model's intrinsic computational processes (e.g., attention), without relying on rule-based reconstruction.
3. **Perturbation-Specific Behaviors**: The effectiveness and model response vary by perturbation type. UCC-Inj is highly disruptive to tokenization yet learnable; models often implicitly filter noise. The Shuffle-Method is remarkably tolerated, likely aided by the local continuity of positional embeddings like RoPE, allowing semantic reassembly despite local order variance. The Typo-Method is most damaging, as it directly alters or removes character content, erasing lexical information and challenging recovery, especially at high perturbation rates.

In summary, this study unveils a previously under-explored layer of robustness in LLMs, highlighting their capacity to perform semantic reasoning even when surface form is heavily corrupted. Our research has broad practical applications, as discussed in Section A.

## 8. Limitation and Future Work

This study has several limitations that suggest directions for future work.

**More high-level tasks.** Our evaluation focuses on low-level robustness within two high-level task settings: mathematics and comprehension. While these domains are representative and sensitive to input corruption, other tasks such as long-context reasoning or multimodal processing may exhibit different robustness patterns. Exploring them could shed further light on the interaction between character-level noise and higher-level capabilities.

**Mechanism formalization.** We identified implicit low-level denoising as a key factor but did not fully formalize this mechanism or trace its origins to specific model components, architectural choices, or training dynamics. A deeper mechanistic analysis, such as using techniques from circuit analysis, could elucidate 'how' and 'when' during forward pass or training this robustness emerges. Linking these behaviors to properties of tokenizers, positional encodings, and attention patterns remains an open challenge.

# Impact Statement

This paper presents work whose goal is to advance the field of machine learning. There are many potential societal consequences of our work, none of which we feel must be specifically highlighted here.

# Acknowledgements

This work was supported by National Natural Science Foundation of China (No. 62506197) and National Key R&D Program of China (2023YFA1009500).

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

# A. Broader Impact of Our Findings

In this section, we will talk about some real scenario applications about our findings in this paper.

**Robustness for Safety-Critical Filtering.**   The observed character-level robustness suggests potential applications in safety-critical content filtering, such as detecting not-safe-for-work (NSFW) or harmful text that has been intentionally obfuscated with perturbations. Models that maintain comprehension under noisy inputs could be more resilient to adversarial attempts to evade filters through character-level manipulations. However, as our experiments show, performance degrades when noise exceeds a certain intensity, indicating that such robustness alone is insufficient for fully reliable detection and must be complemented with other defensive mechanisms.

**Error Correction in OCR and Noisy Text Environments.**   The ability to understand heavily perturbed text implies that LLMs can partially compensate for common optical character recognition (OCR) errors, such as misrecognized characters or insertion/deletion artifacts. This intrinsic denoising capability may facilitate the automatic correction and interpretation of digitized documents, historical texts, or user-generated content where the character set is noisy or partially corrupted, thereby improving the reliability of downstream text-processing applications.

**Improved Handling of Low-Resource and Orthographically Irregular Languages.**   Character-level robustness may also benefit the processing of languages with non-standardized spelling, frequent phonetic variations, or limited lexical resources. By relying less on strict word-level tokenization and more on contextual character patterns, models could better infer meaning from orthographically irregular or informal text. This could enhance performance for low-resource languages where large, clean corpora are scarce, supporting more inclusive and widely applicable natural language processing systems.

**Exploiting Model Vulnerability for Anti-Cheating and Content Protection.**   The relative inrobustness of LLMs to intensive invisible perturbations, such as UCC-3 (three Unicode variation selectors per character), reveals a potential defensive application: intentionally degrading machine readability while preserving human legibility. In scenarios where unauthorized LLM usage should be discouraged—for example, in online examinations or secure text distribution—injecting structured invisible noise could substantially hinder models from accurately parsing and reasoning about the textual content. While not a foolproof barrier, this approach offers a low-cost, deployable mechanism to raise the cost of automated cheating or unauthorized content scraping, complementing other integrity-preserving measures. It underscores that controlled vulnerability can itself be repurposed as a tool for content protection in human-centric interfaces.

# B. Additional Results

## B.1. Model Score on datasets

*Table 7.* Models' score under UCC-Inj

| Model Name | 0-Inj | 1-Inj | 2-Inj | 3-Inj |
|---|---|---|---|---|
| DeepSeek-Reasoner | 0.944 | 0.885 | 0.800 | 0.706 |
| DeepSeek-Chat | 0.963 | 0.822 | 0.610 | 0.328 |
| Qwen3-235B(A22B)Thinking | 0.964 | 0.798 | 0.522 | 0.366 |
| Qwen3-235B(A22B) | 0.948 | 0.663 | 0.343 | 0.199 |
| Qwen3-30B(A3B)Thinking | 0.956 | 0.575 | 0.070 | 0.022 |
| Qwen3-30B(A3B) | 0.941 | 0.434 | 0.047 | 0.026 |
| Llama-3.1-70B | 0.936 | 0.100 | 0.011 | 0.011 |
| Llama-3-70B | 0.905 | 0.114 | 0.014 | 0.008 |
| Qwen3-32B | 0.958 | 0.536 | 0.029 | 0.010 |
| Qwen3-14B | 0.947 | 0.117 | 0.006 | 0.003 |
| Qwen3-8B | 0.932 | 0.188 | 0.006 | 0.007 |
| Qwen3-4B | 0.897 | 0.151 | 0.011 | 0.006 |
| Qwen3-1.7B | 0.774 | 0.005 | 0.005 | 0.003 |
| Qwen3-0.6B | 0.590 | 0.005 | 0.001 | 0.002 |

*Table 8.* Models' score under Typo-Method

| Model Name | 0-Typo | 30-Typo | 50-Typo | 70-Typo | 100-Typo |
|---|---|---|---|---|---|
| Qwen3-30B(A3B) | 0.941 | 0.482 | 0.295 | 0.122 | 0.011 |
| Qwen3-235B(A22B) | 0.948 | 0.503 | 0.387 | 0.200 | 0.033 |
| Qwen3-235B(A22B)Thinking | 0.956 | 0.684 | 0.510 | 0.299 | 0.066 |
| Qwen3-30B(A3B)Thinking | 0.956 | 0.616 | 0.376 | 0.185 | 0.027 |
| DeepSeek-Chat | 0.963 | 0.690 | 0.528 | 0.359 | 0.119 |
| DeepSeek-Reasoner | 0.944 | 0.748 | 0.614 | 0.457 | 0.160 |
| Llama-3.1-70B | 0.936 | 0.625 | 0.425 | 0.237 | 0.069 |
| Llama-3-70B | 0.905 | 0.563 | 0.379 | 0.222 | 0.047 |
| Qwen3-32B | 0.958 | 0.620 | 0.414 | 0.247 | 0.055 |
| Qwen3-14B | 0.947 | 0.540 | 0.318 | 0.124 | 0.008 |
| Qwen3-8B | 0.932 | 0.459 | 0.271 | 0.138 | 0.020 |
| Qwen3-4B | 0.909 | 0.331 | 0.158 | 0.064 | 0.015 |
| Qwen3-1.7B | 0.774 | 0.301 | 0.147 | 0.052 | 0.017 |
| Qwen3-0.6B | 0.590 | 0.172 | 0.066 | 0.023 | 0.006 |

*Table 9.* Models' score under Shuffle-Method

| Model Name | 0-Shuffle | 30-Shuffle | 50-Shuffle | 70-Shuffle | 100-Shuffle |
|---|---|---|---|---|---|
| Qwen3-30B(A3B) | 0.941 | 0.716 | 0.605 | 0.513 | 0.407 |
| Qwen3-30B(A3B)Thinking | 0.956 | 0.815 | 0.729 | 0.612 | 0.384 |
| Qwen3-235B(A22B) | 0.948 | 0.857 | 0.803 | 0.760 | 0.716 |
| Qwen3-235B(A22B)Thinking | 0.964 | 0.918 | 0.879 | 0.882 | 0.580 |
| DeepSeek-Chat | 0.963 | 0.875 | 0.851 | 0.842 | 0.826 |
| DeepSeek-Reasoner | 0.944 | 0.919 | 0.920 | 0.911 | 0.904 |
| Llama-3.1-70B | 0.936 | 0.798 | 0.703 | 0.574 | 0.358 |
| Llama-3-70B | 0.905 | 0.718 | 0.602 | 0.464 | 0.266 |
| Qwen3-32B | 0.958 | 0.830 | 0.771 | 0.707 | 0.622 |
| Qwen3-14B | 0.947 | 0.758 | 0.656 | 0.527 | 0.267 |
| Qwen3-8B | 0.932 | 0.670 | 0.530 | 0.371 | 0.140 |
| Qwen3-4B | 0.909 | 0.575 | 0.407 | 0.258 | 0.071 |
| Qwen3-1.7B | 0.774 | 0.448 | 0.255 | 0.153 | 0.031 |
| Qwen3-0.6B | 0.590 | 0.265 | 0.146 | 0.048 | 0.008 |

## B.2. Categories of Models' Responses on GSM8K

The results are shown in Table 10, Table 11 and Figure 8.

*Table 10.* Percentage of correct/incorrect answers in the three types with different prompt strategies, evaluated with Qwen3-30B (A3B) non-thinking mode on GSM8K under *1-Inj*.

| Prompt Strategy | A | | | B | | | C | | |
|---|---|---|---|---|---|---|---|---|---|
| | correct (c/t) | wrong | total | correct (c/t) | wrong | total | correct (c/t) | wrong | total |
| w.o. instruction | 20.9% (93%) | 1.5% | 22.4% | 12.5% (28%) | 32.4% | 44.9% | 10.9% (33%) | 21.8% | 32.7% |
| w. instruction | 62.7% (76%) | 19.5% | 82.2% | 4.8% (28%) | 12.4% | 17.2% | 0.0% (0%) | 0.5% | 0.5% |
| w. 1-shot | 74.3% (85%) | 13.2% | 87.5% | 4.9% (40%) | 7.5% | 12.4% | 0.0% (0%) | 0.1% | 0.1% |

## B.3. Models' performance under character-level tokenization

The results are shown in Table 12.

*Table 11.* Percentage of correct/incorrect answers in the different proportion of noised words with w.o. instruction, evaluated with Qwen3-30B (A3B) non-thinking mode on GSM8K under Typo-Method and Shuffle-Method.

| Method | A correct (c/t) | wrong | total | B correct (c/t) | wrong | total | C correct (c/t) | wrong | total |
|---|---|---|---|---|---|---|---|---|---|
| *30-Typo* | 47.1% (69.9%) | 20.2% | 67.3% | 1.1% (3.8%) | 29.0% | 30.1% | 0% (0%) | 2.6% | 2.6% |
| *50-Typo* | 27.1% (56.2%) | 21.2% | 48.3% | 2.4% (5.6%) | 39.7% | 42.0% | 0% (0%) | 9.7% | 9.7% |
| *70-Typo* | 10.0% (34.8%) | 18.7% | 28.7% | 2.1% (4.3%) | 47.2% | 49.4% | 0.1% (0.4%) | 21.8% | 21.9% |
| *100-Typo* | 0.5% (3.3%) | 15.7% | 16.2% | 0.6% (1.3%) | 45.0% | 45.6% | 0% (0%) | 38.2% | 38.2% |
| *30-Shuffle* | 69.9% (81.3%) | 16.1% | 86.0% | 1.4% (10.6%) | 12.2% | 13.6% | 0% (0%) | 0.4% | 0.4% |
| *50-Shuffle* | 57.5% (75.6%) | 18.6% | 76.1% | 3.0% (13.2%) | 19.9% | 23.0% | 0% (0%) | 0.9% | 0.9% |
| *70-Shuffle* | 44.8% (66.3%) | 22.7% | 67.6% | 3.6% (11.3%) | 27.9% | 31.5% | 0% (0%) | 1.0% | 1.0% |
| *100-Shuffle* | 29.9% (62.6%) | 17.8% | 47.7% | 8.6% (16.6%) | 43.4% | 52.0% | 0% (0%) | 0.3% | 0.3% |

*Table 12.* GSM8K accuracy with character-level or normal tokenization.

| Model | Character Level | Normal |
|---|---|---|
| Qwen3-32B | 0.930 | 0.956 |
| Qwen3-30B (A3B) | 0.914 | 0.941 |
| Qwen3-14B | 0.915 | 0.946 |
| Qwen3-8B | 0.868 | 0.932 |
| Qwen3-4B | 0.864 | 0.909 |
| Qwen3-1.7B | 0.586 | 0.853 |
| Qwen3-0.6B | 0.240 | 0.589 |
| LLama3.1-70B | 0.883 | 0.936 |
| LLama3.1-8B | 0.696 | 0.828 |

## B.4. Results of Llama in Denoising

The results are shown in Table 13 and Table 14.

*Table 13.* Percentage of correct/incorrect answers in the three types with different prompt strategies, evaluated with Llama3.1-70B on GSM8K under *1-Inj*.

| Prompt Strategy | A correct (c/t) | wrong | total | B correct (c/t) | wrong | total | C correct (c/t) | wrong | total |
|---|---|---|---|---|---|---|---|---|---|
| w.o. instruction | 0.2% (100%) | 0% | 0.2% | % 0.8(17%) | 4.0% | 4.8% | 9.1% (10%) | 86.0% | 95.1% |
| w. instruction | 2.2% (85%) | 0.4% | 2.6% | 4.9% (20%) | 19.2% | 24.1% | 9.3% (13%) | 64.0% | 73.3% |
| w. 1-shot | 40.6% (80%) | 10.3% | 50.9% | 10.8% (22%) | 38.3% | 49.1% | 0.0% (0%) | 0.0% | 0.0% |

*Table 14.* GSM8K accuracy of Llama3.1-70B with different prompt strategies.

| Model | Prompt Strategy | Baseline | *1-Inj* | *2-Inj* | *3-Inj* |
|---|---|---|---|---|---|
| | w.o. instruction | | 0.100 | 0.011 | **0.011** |
| Llama3.1-70B | w. instruction | 0.936 | 0.162 | 0.007 | 0.008 |
| | w. 1-shot | | **0.508** | **0.033** | 0.010 |

## B.5. Visualization of Attention to ASCII Characters

The figure is shown in Figure 9.

## B.6. Result For Linear Probing

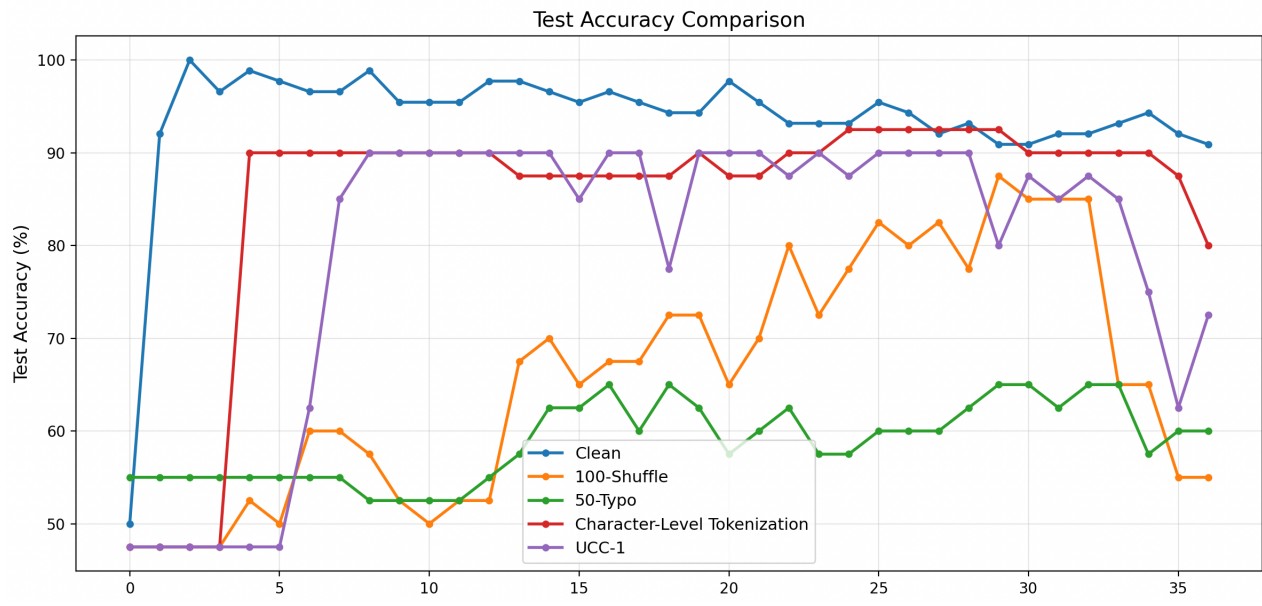

*Figure 10.* Result of linear probing for case 'river'

# C. Prompts

We conducted tests using Chinese prompts on DeepSeek and Qwen, while employing English prompts on Llama. The meaning, structure, and details of both sets of prompts are identical. The English prompts were translated sentence by sentence from the Chinese prompts, ensuring logical coherence and linguistic fluency. The prompts for each experiment are listed in the table below. If an experiment did not involve testing the Llama model, the English version is provided only to illustrate the meaning of the Chinese prompts applied to DeepSeek and Qwen.

## C.1. For GSM8K

*Table 15.* Prompt for evaluate UCC-Inj on GSM8K dataset

| Chinese | English |
|---|---|
| 你要解决这个问题，把答案放在\\boxed{} 里。问题的答案总是整数。\\boxed{} 的内容是评判你答案正确性的唯一依据，它必须仅包含一个整数。例如，如果答案是10000，你需要回答\\boxed{10000}。 回答\\boxed{10,000}、\\boxed{1e4}、\\boxed{$10000$} 等都不得分。题目如下： | I will ask you a problem, and you must answer it. The answer to this problem is always an integer. You must place your answer within \\boxed{}. The content of \\boxed{} is the sole basis for evaluating the correctness of your answer, and it must contain only a single integer. For example, if the answer is 10000, you should respond with \\boxed{10000}. Answers such as \\boxed{10,000}, \\boxed{1e4}, or \\boxed{$10000$} will not be scored. The problem is: |

## C.2. Prompt Models to Explicitly Denoise

*Table 16.* Prompt Models to Explicitly Denoise

| Chinese | English |
|---|---|
| 你要解决这个问题，把答案放在\\boxed{}里。问题的答案总是整数。这个问题的文本是被噪声污染过的，存在很多不可见的无意义字符。你要去掉这些无意义字符，然后理解文本，最后回答问题。\\boxed{}的内容是评判你答案正确性的唯一依据，它必须仅包含一个整数。例如，如果答案是10000，你需要回答\\boxed{10000}。回答\\boxed{10,000}、\\boxed{1e4}、\\boxed{$10000$}等都不得分。题目如下： | I will ask you a problem, and you must answer it. The answer to this problem is always an integer. The text of this problem has been corrupted by noise, containing many invisible and meaningless characters. You need to remove these meaningless characters, then understand the text, and finally answer the question. You must place your answer within \\boxed{}. The content of \\boxed{} is the sole basis for evaluating the correctness of your answer, and it must contain only a single integer. For example, if the answer is 10000, you should respond with \\boxed{10000}. Answers such as \\boxed{10,000}, \\boxed{1e4}, or \\boxed{$10000$} will not be scored. The problem is: |

## C.3. LLM-Assisted Categorization of Model Responses

**We used the Chinese translated version of the following prompt on Qwen-30B(A3B).** I am currently conducting an experiment related to the capability evaluation of large language models. Specifically, I will give the model a question mixed with noise, and the model must correctly understand the question and then answer it.

The model's responses are categorized as follows:

[A] During the response process, the model outputs the correct noise-free text (the meaning should be accurate), and then provides the problem-solving process and answer (regardless of whether the answer is correct).

[B] During the response process, the model attempts to output noise-free text, but the meaning of the denoised text has errors compared to the original text (or lacks key information), and then provides the problem-solving process and answer (regardless of whether the answer is correct).

[C] The model does not explicitly output noise-free text at all, but directly provides the problem-solving process and answer (or only the answer without the process).

Note: Outputting noise-free text refers to outputting the original English text. If a translated text is directly given, it is also considered as case [C].

Below is an example for the input question: "A robe takes 2 bolts of blue fiber and half that much white fiber. How many bolts in total does it take?":

[A] Class response: We need to first remove the meaningless characters from the text to obtain the clear question content. The original text is: `A robe takes 2 bolts of blue fiber and half that much white fiber. How many bolts in total does it take?` [1] The question is translated as: `A robe takes 2 bolts of blue fiber and half that much white fiber. How many bolts in total does it take?` [2] Solution: - Blue fiber: 2 bolts - White fiber: half of 2, which is 1 bolt. Total: 2 + 1 = 3. The answer is: `$$\boxed{3}$$`

[B] Class response: We need to first clean the meaningless characters from the text to obtain a clear sentence. The original text (after removing meaningless characters) is: > `A robot takes 2 balls of blue and a half of a`

---

[1]It was English in the prompt we used

[2]It was Chinese in the prompt we used

*Table 17.* Substitution Mapping Table

| Char | Substitutes | Char | Substitutes |
|------|-------------|------|-------------|
| a | @, 4, q, s, z | A | 4, @, Q, S, Z |
| b | 8, v, n, g | B | 8, V, N, G |
| c | (, x, v, f | C | (, X, V, F |
| d | s, f, e, r | D | S, F, E, R |
| e | 3, w, r, s, d | E | 3, W, R, S, D |
| f | d, g, r, t | F | D, G, R, T |
| g | 9, f, h, t, y | G | 9, F, H, T, Y |
| h | g, j, y, u | H | G, J, Y, U |
| i | 1, !, u, o, j, k | I | 1, !, U, O, J, K |
| j | h, k, u, i | J | H, K, U, I |
| k | j, l, i, o | K | J, L, I, O |
| l | 1, \|, k, ;, o, p | L | 1, \|, K, ;, O, P |
| m | n, ,, j, k | M | N, ,, J, K |
| n | b, m, h, j | N | B, M, H, J |
| o | 0, Q, i, p, k, l | O | 0, Q, I, P, K, L |
| p | o, [, l, ; | P | O, [, L, ; |
| q | 9, Q, w, ' | Q | 9, W, ' |
| r | e, t, d, f | R | E, T, D, F |
| s | 5, $, a, d, z, x | S | 5, $, A, D, Z, X |
| t | 7, +, r, y, f, g | T | 7, +, R, Y, F, G |
| u | y, i, h, j | U | Y, I, H, J |
| v | c, b, f, g | V | C, B, F, G |
| w | q, e, s, a | W | Q, E, S, A |
| x | z, c, s, d | X | Z, C, S, D |
| y | t, u, g, h | Y | T, U, G, H |
| z | 2, a, s, x | Z | 2, A, S, X |

red ball. How many balls in total does it take?[3] Translate and understand the question: A robot takes 2 blue balls and 0.5 red balls. How many balls does it take in total? [4] Calculation process: - Blue balls: 2 - Red balls: 0.5 - Total: 2 + 0.5 = 2.5 balls But the question requires an **integer** answer, so we assess whether there is a misunderstanding in the question. Looking back at the original text: "a half of a red ball" — meaning 0.5 red balls, but the question asks "how many balls", i.e., "how many balls", and "0.5 balls" is not an integer. Therefore, "a half of a red ball" might be a distraction, or the question actually intends to ask "how many whole balls". If so, then the answer is: 2 whole balls. The final answer is: $\boxed{2}$

[C] Class response: A robe requires 2 bolts of blue fiber and half that amount of white fiber, i.e., $ \frac{2}{2} = 1$ bolt of white fiber. Total bolts = 2 (blue) + 1 (white) = 3

I will provide you with the **noise-free** original text and the model's response, and you need to help me classify the model's output into one of the above three categories. Your output should only be a single character, either A, B, or C, without any other content.

## D. Substitution Rule

## E. Models and Hyperparameters

Our experiment are conducted on the following models:

---

[3]It was English in the prompt we used

[4]It was Chinese in the prompt we used

*Table 18.* Experiment models

| Model name | Huggingface Url |
|---|---|
| DeepSeek | https://huggingface.co/deepseek-ai/DeepSeek-V3.1 |
| Qwen3-235B(A22B) | https://huggingface.co/Qwen/Qwen3-235B-A22B |
| Qwen3-30B(A3B) | https://huggingface.co/Qwen/Qwen3-30B-A3B |
| Qwen3-32B | https://huggingface.co/Qwen/Qwen3-32B |
| Qwen3-14B | https://huggingface.co/Qwen/Qwen3-14B |
| Qwen3-8B | https://huggingface.co/Qwen/Qwen3-8B |
| Qwen3-4B | https://huggingface.co/Qwen/Qwen3-4B |
| Qwen3-1.7B | https://huggingface.co/Qwen/Qwen3-1.7B |
| Qwen2.5-32B | https://huggingface.co/Qwen/Qwen2.5-32B |
| Qwen2.5-32B(DeepSeek Distilled) | https://huggingface.co/deepseek-ai/DeepSeek-R1-Distill-Qwen-32B |
| Llama3-8B | https://huggingface.co/meta-llama/Meta-Llama-3-8B |
| Llama3-70B | https://huggingface.co/meta-llama/Meta-Llama-3-70B |
| Llama3.1-8B | https://huggingface.co/meta-llama/Llama-3.1-8B-Instruct |
| Llama3.1-70B | https://huggingface.co/meta-llama/Llama-3.1-70B-Instruct |

Specially, for the DeepSeek and Qwen3-235B(A3B) models, due to their excessively large parameter count, we used the official API instead of running them locally.

We set `do_Sample=False` when generating responses, which means that setting temperature to 0. `max_new_tokens =20000` was set for all experiments(for DeepSeek and Qwen3-235B(A3B), `max_new_tokens` was set by default for offical API).

## F. Other Details of Evaluation

### F.1. Irrelevant prefix

This prefix constructs the following story: Someone is a player who enjoys playing maimai DX (a music game). This player was asked a math question by the staff at the arcade. The content of the math question is the GSM8K problem being tested. We have included an introduction to maimai DX in the prefix, controlling the length of the prefix by adjusting the level of detail in the introduction. The introduction to the game is excerpted from (wikipedia, 2025).

The prefixes are:

**Prefix 1 (short)** Little Fusu is a maimai DX player. She plays maimai DX everyday. Today, fusu was asked a problem by the staff. She must answer it then the staff will turn on the game machine. But fusu is not clever, so you need to help her to solve the problem.

**Prefix 2 (moderate)** Little Fusu is a maimai DX player. She plays maimai DX everyday. Maimai DX is an arcade rhythm game series developed and distributed by Sega, in which the player interacts with objects on a touchscreen and executes dance-like movements. The game supports both single player and multiplayer gameplay with up to 2 players per cab. Today, fusu was asked a problem by the staff. She must answer it then the staff will turn on the game machine. But fusu is not clever, so you need to help her to solve the problem.

**Prefix 3 (long)** Little Fusu is a maimai DX player. She plays maimai DX everyday. Maimai DX is an arcade rhythm game series developed and distributed by Sega, in which the player interacts with objects on a touchscreen and executes dance-like

movements. The game supports both single player and multiplayer gameplay with up to 2 players per cab. The defining feature of Maimai's arcade cabinet is its striking resemblance to a front-loading washing machine. Early advertisements and promotional materials contained different joke catchphrases, such as 'It's not a washing machine!' and 'No water.' At its first deployment test, Maimai had 'No washing' written on its cabinet. Early in development, the original concept was actually a music player instead of a rhythm game. Up to 2 players can play per cabinet, and up to 4 players can play together with more cabinets. The motherboard can be accessed using the back door on the right side. The original design was revealed to have been made in collaboration with Sharp in Sega's Maimai LIVE 2014 Washing Festival event. Today, fusu was asked a problem by the staff. She must answer it then the staff will turn on the game machine. But fusu is not clever, so you need to help her to solve the problem.

## G. LLM Usage Declaration

This work utilized some LLMs to support specific tasks within the research pipeline. The generation of test code for a subset of the experiments was assisted by DeepSeek-Reasoner. To quantitatively analyze the high-level denoising behavior of the models under study, the outputs were classified into three categories using Qwen3-30B (A3B). Finally, the writing of the paper was polished and refined with the aid of both DeepSeek and ChatGPT.

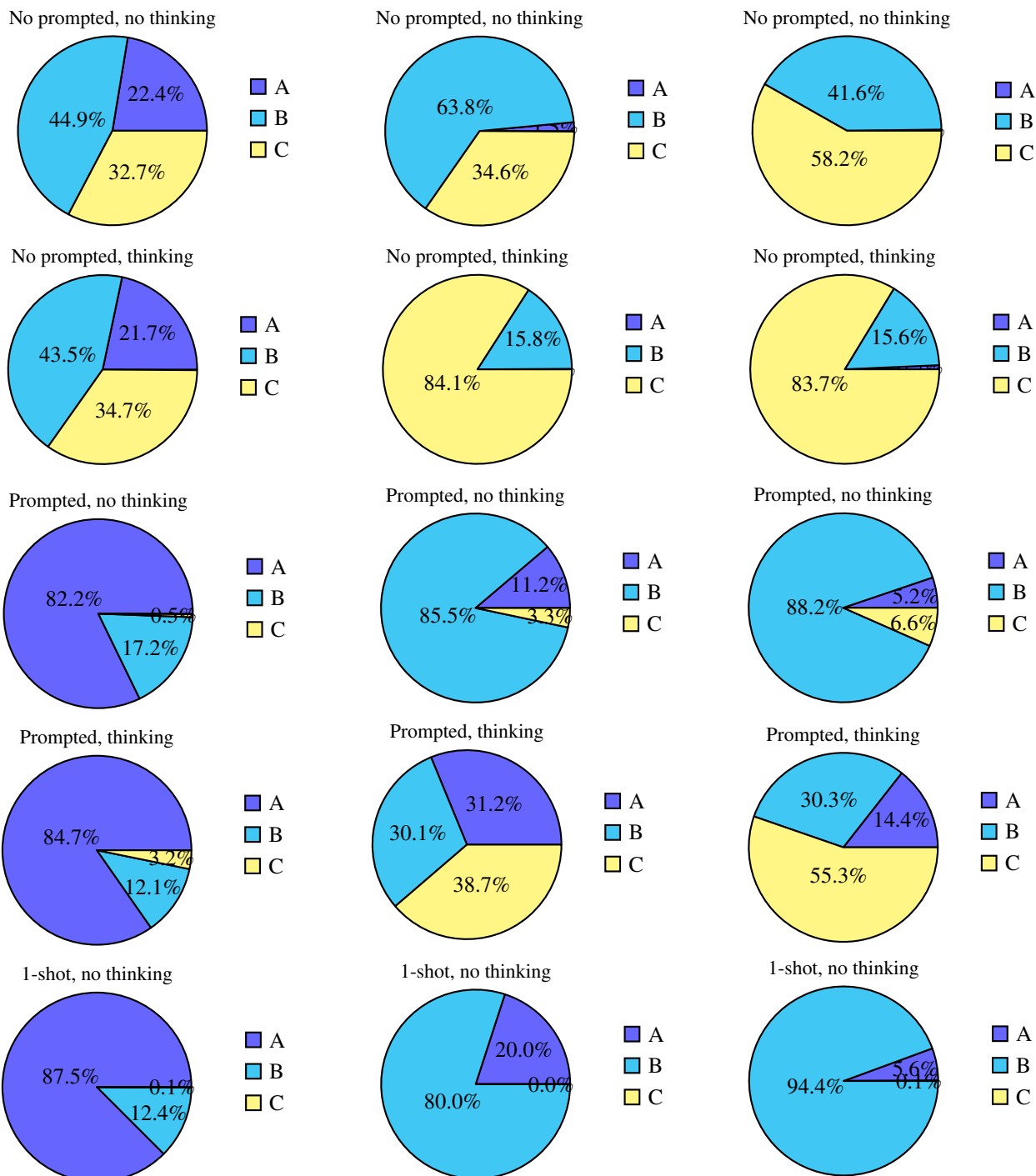

*Figure 8.* The proportions of three categories on Qwen3-30B(A3B). One line from left to right refers to *1-Inj*, *2-Inj*, and *3-Inj*.

Headers indexes

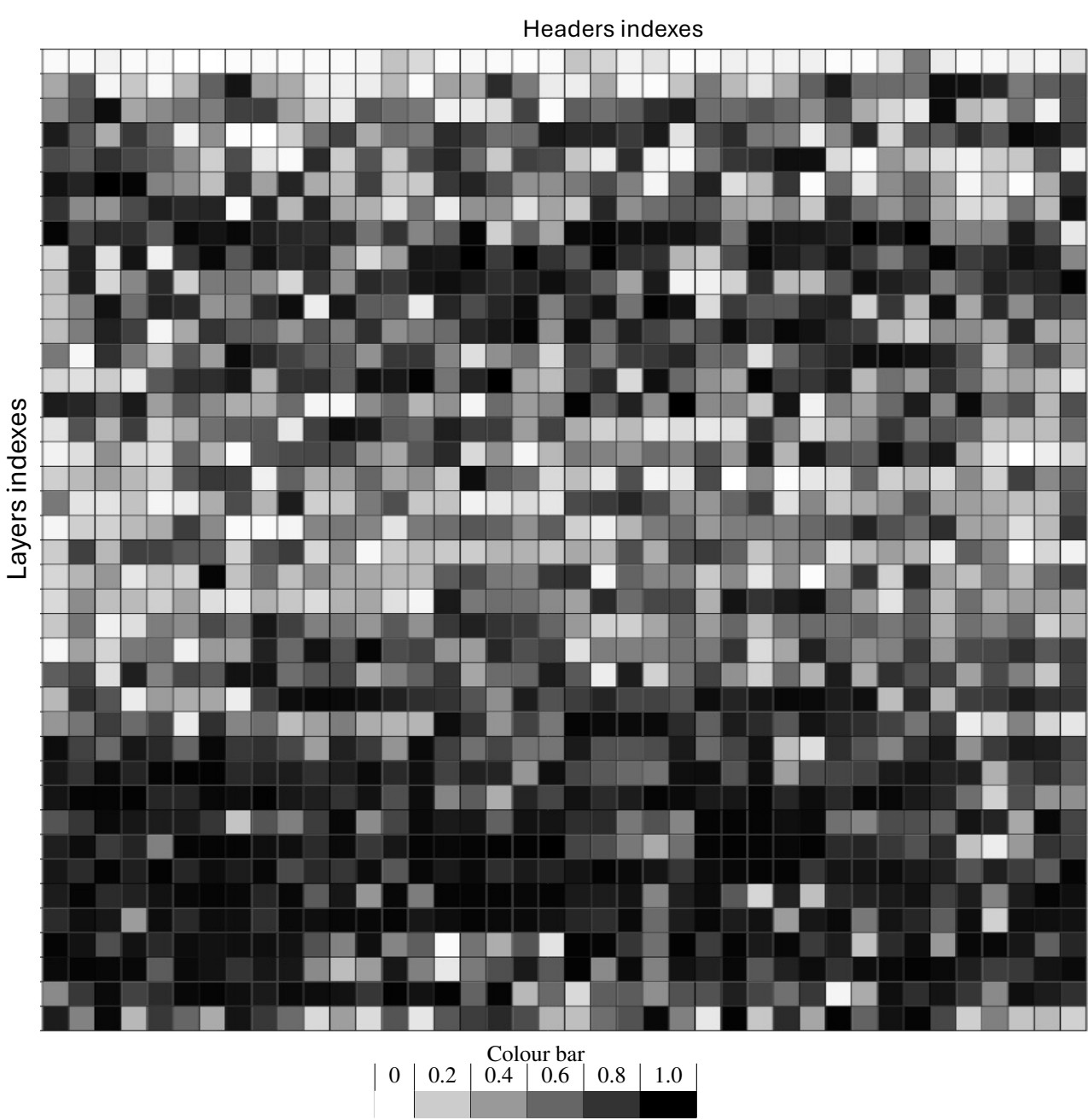

Colour bar

| 0 | 0.2 | 0.4 | 0.6 | 0.8 | 1.0 |

*Figure 9.* Attention of Headers to ASCII Characters of Qwen3-14B under *3-Inj*. From top to bottom, each row represents the heads in each layer, and the darker the color of each cell, the higher the attention to ASCII characters.

