# OpenReview forum: "Understanding the Ability of LLMs to Handle Character-Level Perturbation"
_ICML.cc/2026/Conference — ICML 2026 regular_

### Official Review · Reviewer_xX1Q · 2026-03-01

**Soundness:** 3
**Presentation:** 3
**Significance:** 2
**Originality:** 3
**Overall Recommendation:** 5
**Confidence:** 3

**Summary:**

The authors perform an extensive study of LLMs' ability to recover the meaning of a prompt that has been corrupted in various ways, such as inserting typos, shuffling characters, and injecting invisible unicode characters. They study this character-level perturbation robustness on prompts from GSM8K, finding extensive results.

**Compliance With Llm Reviewing Policy:**

Affirmed.

**Final Justification:**

Overall I don't see a reason to reject this paper, other than perhaps a lack of clear narrative which I trust the authors will address (I guess it can be hard with these kinds of result-heavy papers). In the spirit of decisiveness, I'm raising my score to a 5.

**Key Questions For Authors:**

## Motivation and storytelling

Clearly a lot of work went into the experiments in this paper (and clearly there are many more experiments that could be done). What is the motivation behind this work? Do you have a specific threat model (whether intentional like jailbreaks, or unintentional like OCR) in mind? Based on your findings, overall, do you have a recommendation for how to make LLMs more robust to character-level perturbations in different settings? I understand that this flavor of paper is more about getting lots of insights and sweeping recommendations might be hard to make, but having a stronger narrative from introduction through experiments and results and conclusion would significantly strengthen it (and help keep the reader attentive through all the results).

## Settings

* In Section 8 (line 420 right column), you mention having studied two settings, code generation and mathematics. But in Section 3 you only mention GSM8K, a math benchmark. Did you in fact have code generation experiments too?

## Nits

* Table 3 is really large, consider standardizing table font size
* line 316 left column, what do "6 and 7" refer to?
* line 292 right column, what does "stat the proportion" mean?
* line 356 right column, talking about S5.2.1 right after S5.2.2 finishes feels like a jump

**Limitations:**

yes

**Strengths And Weaknesses:**

## Soundness

Paper appears technically sound

## Presentation

Paper is well presented overall. A bit of a slog to read through, but that's how it is with these highly empirical result-heavy works.

## Significance

Results are not groundbreaking in themselves, but the overall direction of study is compelling and worthwhile

## Originality

While studying character-level robustness is a natural idea, it's original to have a work so fully focused on it.

---

> ### Author Rebuttal · Authors · 2026-03-31
>
> We sincerely thank the reviewer for the positive assessment and constructive feedback. We are greatly encouraged that you found our focus on character-level robustness to be an original, compelling, and worthwhile direction of study. We also deeply appreciate your careful reading, which helped us identify several lingering typos. We have addressed your insightful questions below.
>
> **On Motivation and Storytelling:**
> As you rightly pointed out, our primary motivation is to reveal and explain the mechanisms underlying models' character-level robustness. Regarding specific threat models, we consider both unintentional errors (such as OCR artifacts and human typos) and intentional malicious attacks (such as adversarial noise injection). Highlighting the models' intrinsic error-correction and defensive capabilities under these conditions is practically meaningful. We have provided a more detailed discussion of these specific application scenarios in Appendix A. Ultimately, we believe that achieving a deeper understanding of these inherent robustness mechanisms is the critical first step toward designing principled methods to further enhance them. We will revise our introduction and conclusion to weave this narrative more strongly throughout the paper, as you suggested.
>
> **On the "Code Generation" Setting:**
> Thank you for catching this discrepancy in Section 8. In an earlier version of this work, we indeed conducted experiments on code generation tasks (using HumanEval and an algorithmic competition dataset), and the results were highly consistent with our findings on GSM8K. However, as we expanded our mechanistic experiments, we had to remove the code generation section due to space constraints and to maintain a more focused narrative. We inadvertently left the mention of code generation in the limitations section. We will remove this phrase in the revised manuscript.
>
> **On Nits and Presentation:**
> We sincerely apologize for the typographical errors and structural jumps. To clarify:
> * "6 and 7"  refers to **Figure 6** and **Figure 7**.
> * "stat the proportion"  was a typo and should read "**calculated the proportion**".
>
> Thank you for your suggestions about the paper structure and format. We will adjust our paper according to your suggestions.
>
> Thank you again for helping us improve the presentation of our work.

---

> > ### Author Rebuttal · Reviewer_xX1Q · 2026-04-04
> >
> > Thank you for your responses, this answers all my questions. Best of luck with the submission!

---

> > > ### Author Response · Authors · 2026-04-05
> > >
> > > Thank you again for your kind words and support. Your valuable comments helped us a lot in improving our work. Best regards!

---

### Official Review · Reviewer_gJ19 · 2026-03-08

**Soundness:** 2
**Presentation:** 2
**Significance:** 3
**Originality:** 2
**Overall Recommendation:** 3
**Confidence:** 4

**Summary:**

This paper studies how large language models perform under three main types of character-level perturbations: (1) typo-like corruption, including deletion, insertion, adjacent character swaps, and substitution; (2) within-word character shuffling; and (3) invisible Unicode character injection. Evaluating on GSM8K math word problems, the authors find that many LLMs remain surprisingly robust even when the input becomes heavily corrupted. They further analyze how robustness varies with model family, model size, prompt length, and perturbation subtype, finding in particular that larger models are generally more robust.

The paper then investigates possible explanations for this robustness. First, it shows that models already handle abnormal tokenization surprisingly well, including character-level tokenization and random in-word segmentation. Second, it examines whether models explicitly rewrite corrupted inputs in their chain-of-thought. While explicit rewriting can improve performance, especially with prompting, the authors argue that the main source of robustness is an implicit internal denoising process rather than rewriting alone.

**Compliance With Llm Reviewing Policy:**

Affirmed.

**Final Justification:**

The authors have addressed most of my concerns, and I appreciate the added SQuAD results and probing analysis. However, I remain unconvinced that the evaluation is broad enough to justify the paper’s central claims. As a result, I am raising my score only modestly, from 2 to 3.

**Key Questions For Authors:**

1. Can you clarify the perturbation severity used in Section 4.3 for the individual typo operations? It would be much easier to interpret the comparison between deletion, insertion, substitution, and swapping if the degree of corruption were normalized or reported explicitly, for example using edit distance.

2. In the statement-length experiment, are the longer statements also perturbed, or are only the original problem statements noised? Relatedly, could you comment on the unusually low Baseline Prefix2 result and confirm whether this is expected rather than a typo or implementation issue?

3. In the visible-character replacement experiment, why did you only test replacing variation selectors with the character "A"? It seems plausible that the effect could depend on which visible character is inserted.

4. Could you better justify the paper's framing around "mechanisms" of robustness? Much of Section 5 reads as behavioral analysis rather than mechanistic analysis in the interpretability sense, and the discussion of RoPE as a possible explanation for shuffling robustness feels weak.

**Limitations:**

yes

**Strengths And Weaknesses:**

## Soundness
The paper is generally technically solid in its core empirical setup, and the abstract and introduction do a good job motivating the problem and clearly defining the three perturbation types. The perturbation methods themselves are easy to understand, and Figure 1 is especially effective in making them concrete. The experiments are broad enough to establish the main behavioral result: many LLMs retain non-trivial performance under character-level perturbations, and robustness varies across perturbation type and model size.

That said, I have several concerns about how strongly some of the paper's claims are supported. First, the heavy reliance on GSM8K may be misleading, since GSM8K is a well-known contaminated benchmark. If some models have memorized parts of the benchmark, then robustness to perturbed inputs may partly reflect answer memorization rather than genuine robustness to corrupted text. This does not invalidate the results, but it weakens the strength of the central claim unless the authors can rule this out with additional benchmarks (preferably, new or creative benchmarks). Second, some experimental details are insufficiently specified. In Section 4.3, when individual typo operations are studied, it is unclear how much perturbation is applied in each case; measuring and reporting perturbation extent using edit distance would make the comparison more principled. In the statement-length experiment, it is also unclear whether the statements themselves are noised, and the very low Baseline Prefix2 result looks suspicious enough that it may reflect a typo or implementation issue. Similarly, in the visible-character replacement experiment, it is interesting that inserting "A" is highly disruptive, but it is not clear why only "A" was tested rather than a broader set of alphabetic or visible characters.

I also think the paper overstates its analysis of mechanisms. Section 5 provides some useful behavioral evidence, and the attention visualization for invisible-character perturbations is interesting, but overall the work is much more behavioral than mechanistic. The discussion of shuffling robustness possibly arising from RoPE is especially underdeveloped. If the paper wants to claim insight into "mechanisms," I would expect a more principled analysis in the style of interpretability work rather than mainly prompt/output-level behavior and limited attention plots. The authors do partly acknowledge this in the limitations, which is good, but the framing in the main text should be revised.

## Presentation
The paper has some clear strengths in presentation. The abstract and introduction are very well written, the setup is easy to follow, and the three perturbation types are explained clearly. Figure 1 is particularly strong. One concrete improvement would be to end the introduction with a few headline quantitative results so the reader leaves the introduction with a better sense of the main empirical findings.

I take more issues with everything after the introduction. I strongly think the related work section should appear before the method section rather than as a short Section 6 at the end. There are several closely related papers and benchmarks on this topic, including [CUTE](https://aclanthology.org/2024.emnlp-main.177/) and [Huang et al. 2023](https://aclanthology.org/2023.findings-acl.770/), and the current related work discussion is too brief to properly situate the paper. Moving related work earlier would give the authors a natural place to explain what is genuinely new here versus prior benchmarks. There is also relevant work in memorization that may bear directly on the interpretation of results, such as [Liu et al. 2025](https://arxiv.org/abs/2503.17514), and that should be discussed as well. There's also the [Broken Tokens](https://arxiv.org/abs/2506.19004) paper that appears to be very closely related to this work, and should be discussed.

I also found the presentation of results weaker than it should be. The prose of Sections 4, 5, and 6 is fine, but the placement and design of figures and tables could be improved. Figures 5, 6, and 7 contain a lot of repetitive information and could likely be compressed into a single figure or otherwise streamlined. More importantly, the paper relies too heavily on figures and tables in the appendix. Reading the paper should not require much back-and-forth between the main text and appendix, as this hurts readability. Some of the most interesting material, such as the attention visualization, are in the appendix.

## Significance
The paper addresses an important and practical question: how robust are modern LLMs to character-level corruption, including forms of noise that break human readability or standard tokenization? This is a relevant problem for both deployment and scientific understanding. On the practical side, robustness to OCR noise, user typos, and Unicode-based corruption clearly matters. On the scientific side, understanding why models remain capable under these perturbations can help clarify what information they rely on and how brittle or flexible their processing actually is.

My issue is that the paper's significance is somewhat limited by the benchmark choice and the gap between the framing and the actual analysis. If the central evidence comes solely from GSM8K, then concerns about contamination narrow how broadly we should interpret the findings. And if the paper frames itself as investigating mechanisms but mostly provides behavioral observations, then the contribution is more modest than the framing suggests. I still think the problem is significant, but the paper would be stronger if it either broadened the empirical evaluations or narrowed its claims.

## Originality
The paper has some meaningful originality, though more in the form of limited new empirical evidence than in introducing a fundamentally new method. The combination of perturbation types, especially the inclusion of invisible Unicode character injection alongside more standard typo and shuffle perturbations, is interesting and useful.

That said, the paper needs to do a much better job distinguishing itself from prior work. As written, the related work section is too weak to support the novelty claims. There are already benchmarks and papers on character-level robustness, such as CUTE and Huang et al. (2023), and the paper should clearly articulate what is new beyond those papers.

I would describe the paper as moderately original: it does not seem to introduce a radically new task or method, but it does provide a useful empirical combination of ideas and some interesting observations that could deepen understanding of LLM robustness. The originality is real, but it is not yet justified as clearly as it needs to be.

---

> ### Author Rebuttal · Authors · 2026-03-31
>
> We sincerely thank you for the thorough and constructive feedback. We are encouraged that you found our research question well-motivated , the perturbation types informative , and our decomposition of robustness into tokenization and denoising to be an interesting contribution.  Your suggestions have significantly strengthened the paper. We address your concerns below:
>
> **1. Perturbation Severity in Typo Operations**. For individual typo perturbations, each targeted word undergoes **exactly one operation**. Therefore, the edit distance is exactly 1 for all perturbations but swapping. Swapping also occurs exactly once per word.
>
> **2. Statement-Length Experiment & Baseline Typo**. **Yes, the longer prefixes were also perturbed**. We are sorry that reporting the `Baseline Prefix2` as 0.035 was a typographical error; the correct accuracy is **0.935**. We will correct this in the manuscript.
>
> **3. Visible-Character Replacement ('A' vs. Multiple Characters)**. We used a single character ('A') to **maintain a recognizable pattern for humans**, aligning with the premise of UCC-Inj (where noise is invisible, keeping text human-readable). Inserting diverse visible characters makes the text almost illegible. What's more, **we did test this on Qwen3-30B(A3B)**: using subsets of visible ASCII punctuation (sizes 8 and 27) under 1-INJ dropped accuracy to 22.3% and 7.3%, respectively. In contrast, using 256 invisible VS characters maintained 43.4% accuracy. We did not report this result in the paper due to space constraints.
>
> **4. Mechanisms and the RoPE Hypothesis**. We agree that our discussion attributing Shuffle Method performance to RoPE lacked empirical support. Since it is currently infeasible to find a non-RoPE model with comparable capabilities to establish a controlled baseline, **we will remove the speculative discussion** from the paper.  Meanwhile, we agree that a deeper mechanistic explanation is needed. At the mechanistic level, we have **supplemented our study with an additional experiment**:
>
> Using a noun with clear meaning(e.g., "river"), we generated two sentence sets—one ending with the target word, the other with a related word (e.g., "bay", "lake"). We split data 8:2 for training/testing. For each layer of Qwen3-8B, we trained a linear probe (logistic regression) to classify whether the hidden state encodes the target word. Perturbations were applied to the test set, and probes were used during forward passes.
> We hypothesized that the model internally denoises across layers: low accuracy in shallow layers, then a rise in middle-to-late layers as information is restored. Results support this:
> - **Baseline** (unperturbed): ~90% accuracy across all layers.
> - **100-Shuffle**: 50%in shallow layers, then rises significantly to ~80% in middle-late layers, dropping below 70% in final layers.
> - **50-Typo**: similar trend but less pronounced.
> - **Character-level tokenization**: accuracy rises at layer 3 and stays high.
> - **1-Inj**: similar to character-level, but recovery starts later (layer 6).
>
> For different cases, the accuracy curves of the classifier show some variation; nevertheless, they consistently adhere to the pattern of low initial accuracy, a subsequent increase in the middle-to-late layers, and a final decline. This experiment reveals that an internal mechanism capable of autonomously completing the denoising process indeed exists within the model.
>
> Here is the visual plot of our case study "river": ![](https://ibb.co/h1xT3Qkm)
>
> **5. Narrow Scope & GSM8K Contamination**. We evaluated our methods on a random subset of 1,000 samples from the SQuAD dev set. The results perfectly **align with our initial findings**: robustness persists and degrades as perturbatio increases, as described below. Regarding GSM8K contamination, potential memorization does not invalidate our core conclusion. We are evaluating the model's low-level ability to parse corrupted text, not the math reasoning. Even if a model has memorized an answer, it must still successfully "read" and decode the perturbed information to trigger that memorization.
>
> | Metrics | Baseline | 30-Typo | 50-Typo_50 | 70-Typo | 100-Typo |
> |:-:|:-:|:-:|:-:|:-:|:-:|
> | em | 63.2 | 30.67 | 17.27 | 5.6 | 0.33 |
> | f1 | 81.48 | 51.18 | 33.31 | 13.25 | 1.39 |
>
> | Metrics | Baseline | 30-Shuffle | 50-Shuffle | 70-Shuffle | 100-Shuffle |
> |:-:|:-:|:-:|:-:|:-:|:-:|
> | em | 63.2 | 35.2 | 24.87 | 14.0 | 6.07 |
> | f1 | 81.48 | 56.26 | 42.84 | 27.16 | 11.18 |
>
> | Metrics | Baseline | 1-INJ | 2-INJ | 3-INJ |
> |:-:|:-:|:-:|:-:|:-:|
> | em | 63.2 | 7.67 | 0.73 | 0.2 |
> | f1 | 81.48 | 9.23 | 1.02 | 0.39 |
>
> **6. Related Work & Presentation**. Thanks for pointing out more related works. We will incorporate these into the Related Work and move the section up. Compared to these works, our study applies a stronger perturbation and focuses primarily on the model's inference-time denoising behaviors. We will also carefully refine our writing and streamline the figures.

---

> > ### Author Rebuttal · Reviewer_gJ19 · 2026-04-02
> >
> > I thank the authors for their rebuttal.
> >
> > The authors have addressed most of my concerns, and I appreciate the added SQuAD results and probing analysis. However, I remain unconvinced that the evaluation is broad enough to justify the paper’s central claims. As a result, I am raising my score only modestly, from 2 to 3.

---

> > > ### Author Response · Authors · 2026-04-05
> > >
> > > We sincerely thank you for your swift response and positive feedback! We appreciate your valuable comments, which have helped us improve the work. We will show the results of the SQuAD evaluation on more models and more case studies for the linear probe experiment in the final version to make our work more convincing. Thank you again for help to improve our work!

---

### Official Review · Reviewer_Aq5D · 2026-03-13

**Soundness:** 3
**Presentation:** 2
**Significance:** 2
**Originality:** 2
**Overall Recommendation:** 4
**Confidence:** 3

**Summary:**

This paper investigates how well LLMs handle three types of character-level perturbations: typo-like errors (Typo Method), within-word character shuffling (Shuffle Method), and invisible Unicode control character injection (UCC-Inj). The authors evaluate on GSM8K across several model families (DeepSeek, Qwen3, Llama) and find that LLMs retain surprising performance even under severe perturbation. They then probe the mechanisms behind this robustness, identifying a dual-layer process: explicit rewriting in chain-of-thought, and implicit low-level denoising.

**Compliance With Llm Reviewing Policy:**

Affirmed.

**Final Justification:**

The rebuttal addressed my two most actionable concerns. The SQuAD evaluation showed that the robustness phenomenon generalizes to reading comprehension, reducing my primary worry about narrow evaluation scope. The linear probe experiment provides meaningful new mechanistic evidence. the progressive layerwise recovery pattern supports the implicit denoising claim far more concretely than the attention visualizations in the original submission. The decision to remove the unsupported RoPE speculation also improves the paper's rigor.

That said, residual concerns remain. The mechanistic probing is limited to a single-word case study and needs broader validation. The model comparison still conflates architecture, tokenizer, and training data effects, which the authors reasonably acknowledge but cannot resolve. The writing requires substantial revision. And while two benchmarks are better than one, the evaluation scope remains modest relative to the generality of the claims.

Weighing these factors, I raise my score from 3 to 4. The core observations are interesting, the rebuttal demonstrated good engagement, and the new experiments move the paper in the right direction. However, the mechanistic contribution still feels preliminary rather than mature, and I believe the paper would benefit from another revision cycle.

**Key Questions For Authors:**

1 Could you run even a small-scale experiment on a non-math task to test generalization?
2 For the RoPE hypothesis, have you considered comparing against a model using absolute positional embeddings?

**Limitations:**

Partially adequate. The authors make a good-faith effort but would benefit from a more candid discussion of misuse potential and fairness considerations.

**Strengths And Weaknesses:**

Strengths

- The core research question is well motivated and practically relevant. Understanding how LLMs handle corrupted inputs matters for OCR pipelines, adversarial robustness, etc. The paper covers a nice range of perturbation types that differ meaningfully in their properties - whether they destroy character information, fragment tokenization, or introduce noise,  and the comparison across these dimensions is informative.
- The decomposition of robustness into tokenization handling and denoising ability is the most interesting contribution. The experiments showing that LLMs handle character-level tokenization with minimal degradation and random in-word segmentation nearly perfectly  are striking and useful findings. The analysis of explicit rewriting behavior (categories A/B/C) and its correlation with accuracy is a nice methodological choice that provides genuine insight into how models process corrupted text.
- The observation about RoPE potentially explaining shuffle robustness is an intriguing hypothesis - nearby positions map to similar embeddings, so shuffled characters within a word remain "close" in embedding space, preserving some bag of characters signal.

Weaknesses
- Narrow evaluation scope- The entire empirical eval uses a single benchmark (GSM8K). Math problems have a particular structure: relatively short statements, constrained vocabulary, numerical answers that may make them unusually amenable to recovery from perturbation. The authors acknowledge this in the limitations, but to me it substantially weakens the generality of their claims. Even one additional task type (like freeform generation, coding, etc) would meaningfully strengthen the paper.

- The RoPE hypothesis is underspecified and untested: The claim that RoPE's local continuity explains Shuffle Method robustness  is presented speculatively without any controlled experiment. One could test this by comparing models with different positional encoding schemes, or by ablating position information. As stated, it's plausible but unsubstantiated.

- Incomplete mechanistic analysis- The paper identifies that implicit denoising happens but doesn't make much progress in explaining how. The attention visualization shows that some heads attend more to ASCII characters under UCC-Inj, but this is a fairly surface-level observation. The cosine similarity tracking is more interesting but also inconclusive. The paper would benefit from more rigorous probing or intervention experiments.

- Possible confounding factors in the model comparison? Different model families differ in architecture, training data, tokenizer, and training procedure. When the paper finds that Qwen3 is more robust than Llama, it's unclear what drives this. Is it the tokenizer vocabulary? Training data composition (e.g., more noisy text)? Model architecture? The paper doesn't attempt to disentangle these factors.

- Writing quality and presentation: The paper has numerous grammatical errors and awkward phrasings throughout (e.g., "donated by 30-Typo" should be "denoted," "we stat the proportion" should be "we state," "find that find different model families"). The organization is reasonable but the paper reads as rushed in places.

---

> ### Author Rebuttal · Authors · 2026-03-31
>
> We sincerely thank you for the constructive feedback and for recognizing the originality and practical relevance of our research question. We are greatly encouraged that you found our decomposition of robustness into tokenization handling and denoising ability to be an interesting and useful contribution. We have carefully addressed your concerns below and added a new mechanistic experiment to strengthen our claims.
>
> **On the narrow evaluation scope**, to address the concern that GSM8K’s specific structure might uniquely aid text recovery, we extended our evaluation to the **[SQuAD](https://huggingface.co/datasets/rajpurkar/squad)**,  which is a reading comprehension dataset. The **results align with our initial findings**: the robustness persists, and performance metrics decrease smoothly as perturbation intensity increases. Notably, models still demonstrate a strong ability to handle the Shuffle Method in this reading comprehension setting. The result confirms that the phenomenon generalizes beyond math word problems.
> The result on SQuAD is shown in the table. Due to the limited time during the rebuttal period, we evaluated our methods on a randomly sampled subset of 1,000 examples from the SQuAD dev set.
> | Metrics | Baseline | 30-Typo | 50-Typo_50 | 70-Typo | 100-Typo |
> |:-:|:-:|:-:|:-:|:-:|:-:|
> | em | 63.2 | 30.67 | 17.27 | 5.6 | 0.33 |
> | f1 | 81.48 | 51.18 | 33.31 | 13.25 | 1.39 |
>
>
> | Metrics | Baseline | 30-Shuffle | 50-Shuffle | 70-Shuffle | 100-Shuffle |
> |:-:|:-:|:-:|:-:|:-:|:-:|
> | em | 63.2 | 35.2 | 24.87 | 14.0 | 6.07 |
> | f1 | 81.48 | 56.26 | 42.84 | 27.16 | 11.18 |
>
>
> | Metrics | Baseline | 1-INJ | 2-INJ | 3-INJ |
> |:-:|:-:|:-:|:-:|:-:|
> | em | 63.2 | 7.67 | 0.73 | 0.2 |
> | f1 | 81.48 | 9.23 | 1.02 | 0.39 |
>
> **On the RoPE hypothesis**, we agree that our discussion attributing Shuffle Method performance to RoPE lacked empirical support. Since it is currently infeasible to find a non-RoPE model with comparable capabilities to establish a controlled baseline, **we will remove the speculative discussion** from the paper. At mechanism level, we conduct a new experiment to export the internal mechanism on denoing, as descriped below.
>
> **On the incomplete mechanistic analysis**,  we agree that a deeper mechanistic explanation is needed. To address this,  at the mechanistic level, we have **supplemented our study with an additional experiment**:
> Using a noun with clear meaning(e.g., "river"), we generated two sentence sets—one ending with the target word, the other with a related word (e.g., "bay", "lake"). We split data 8:2 for training/testing. For each layer of Qwen3-8B, we trained a linear probe (logistic regression) to classify whether the hidden state encodes the target word. Perturbations were applied to the test set, and probes were used during forward passes.
> We hypothesized that the model internally denoises across layers: low accuracy in shallow layers, then a rise in middle-to-late layers as information is restored. Results support this:
> - **Baseline** (unperturbed): ~90% accuracy across all layers.
> - **100-Shuffle**: 50%in shallow layers, then rises significantly to ~80% in middle-late layers, dropping below 70% in final layers.
> - **50-Typo**: similar trend but less pronounced.
> - **Character-level tokenization**: accuracy rises at layer 3 and stays high.
> - **1-Inj**: similar to character-level, but recovery starts later (layer 6).
>
> For different cases, the accuracy curves of the classifier show some variation; nevertheless, they consistently adhere to the pattern of low initial accuracy, a subsequent increase in the middle-to-late layers, and a final decline. This experiment reveals that an internal mechanism capable of autonomously completing the denoising process indeed exists within the model.
>
> Here is the visual plot of our case study "river": ![](https://ibb.co/h1xT3Qkm)
>
> **On possible confounding factors in the model comparison**, we completely agree that differences in architecture, tokenizer vocabulary, and training data composition could all contribute to the observed robustness gap between Qwen3 and Llama. Unfortunately, because the pre-training data and specific training procedures for these models remain closed, we do not have a reliable method to perfectly disentangle these factors. We will explicitly state this limitation in the revised manuscript to ensure readers interpret the model comparisons with the appropriate caveats.
>
> **On writing quality and presentation**,  we sincerely thank you for pointing out the grammatical errors and awkward phrasings. We will carefully proofread the entire manuscript to correct these issues and improve the overall flow and presentation of the paper.

---

> > ### Author Rebuttal · Reviewer_Aq5D · 2026-04-02
> >
> > I thank the authors for their thorough and constructive rebuttal. The new experiments meaningfully address several of our concerns.
> >
> > **SQuAD evaluation.** The addition of a reading comprehension task is the most important change, as it directly addresses the primary concern about generalizability. The results are encouraging. I would ask the authors to clarify which model was used for this evaluation and to include results for at least one additional model in the final version if possible, since cross-model generalization was a key part of the original contribution.
> >
> > **Linear probe experiment.** This is a very interesting addition and represents the kind of mechanistic evidence I felt was missing. That said, we note this is currently a single-word case study. As it stands, it is suggestive rather than definitive, but a strong step in the right direction.
> >
> > **Updated assessment.** In light of the new evidence, particularly the SQuAD results and the probing experiment, I am willing to raise my score from 3 to 4. The paper's core contributions are now better supported, though I still feel the mechanistic analysis would benefit from further development beyond a case study, and the writing requires careful revision.

---

> > > ### Author Response · Authors · 2026-04-05
> > >
> > > We sincerely thank you for your swift response and positive feedback!
> > >
> > > We clarify that we use **Qwen3-30B(A3B)** model (with /nothink mode) for SQuAD evaluation. Thank you for pointing out that we missed clarifying the model name in the rebuttal. We will show the result of the SQuAD evaluation on more models in the final version of the paper.
> > >
> > > For linear probe experiment, we acknowledge that a single case study is not particularly convincing. We will show the probe result on more cases in the final version of the paper.
> > >
> > > Thank you again for your valuable feedback on improving our work!

---

### Official Review · Reviewer_txk4 · 2026-03-14

**Soundness:** 2
**Presentation:** 2
**Significance:** 2
**Originality:** 2
**Overall Recommendation:** 2
**Confidence:** 3

**Summary:**

This paper examines whether modern LLMs can still understand text when the input is heavily corrupted at the character level, and it also tries to explain why that happens. The authors study three perturbation methods, Typo Method, Shuffle Method, and UCC Inj, and show that some recent models still perform surprisingly well even when the text becomes hard to read or tokenization is severely disrupted. They interpret this as evidence that LLMs have a kind of low level robustness. The main experiments are conducted on GSM8K using models from the DeepSeek, Qwen3, and Llama families. Overall, all three perturbation methods reduce performance, but the results under Shuffle Method are especially striking. Some larger models retain much higher accuracy than one might expect. The paper also includes several additional analyses, including comparisons across model families and model sizes, as well as the effects of input length, Unicode versus set size, and visible versus invisible noise. To explain these findings, the authors suggest two factors. One is robustness to abnormal tokenization. The other is a denoising ability that helps the model recover semantic meaning from corrupted character sequences. To support this claim, they run additional experiments in which clean inputs are artificially converted into character level tokenization or randomly segmented in the middle of words. They also analyze cases where the model appears to rewrite corrupted inputs into cleaner forms during reasoning, which they call explicit rewriting. The authors argue that this behavior is associated with higher answer accuracy.

**Compliance With Llm Reviewing Policy:**

Affirmed.

**Key Questions For Authors:**

.

**Limitations:**

Yes

**Strengths And Weaknesses:**

Strengths:

1. The main question of the paper is interesting, and the phenomenon itself is genuinely surprising. Most previous work on character level perturbations has focused on adversarial settings and on how these perturbations can break models. This paper asks the opposite question, namely why some models remain robust even under severe corruption. That framing feels both original and worthwhile. It is also helpful that the paper compares several different perturbation types within a unified setup.

2. The empirical results are also impressive. It is particularly interesting that some models retain substantial accuracy under shuffled text or highly unusual segmentation, even when the input looks almost unreadable to a human. The paper also goes beyond a single benchmark result. It compares model families and model scales, and it further explores factors such as prefix length, VS set size, and whether the perturbation is visually noticeable.

3. Another positive aspect is that the paper does not simply report the phenomenon. It also makes a real effort to explain it. The character level tokenization and random segmentation experiments help isolate the role of abnormal tokenization, and the explicit rewriting analysis provides at least some evidence that denoising behavior may matter for downstream performance. I do not think the mechanism is fully established yet, but I still appreciate that the paper tries to move beyond observation and offer an explanation.

Weakness:

1.The evaluation scope is fairly narrow. Most of the paper’s main evidence comes from GSM8K, so it is difficult to know how broadly the conclusions extend beyond this setting. Math word problems may be unusually recoverable even when the surface form is corrupted, since they often contain recurring structures, familiar templates, and salient numerical cues. Because of this, I think the current experiments support a more limited claim than the paper sometimes suggests.

2. mechanistic analysis is suggestive, but not yet decisive. The paper proposes a two part explanation based on robustness to abnormal tokenization and denoising, but the evidence is still largely behavioral. The observation that explicit rewriting is associated with better performance is interesting, but it does not establish whether rewriting actually improves performance or simply appears more often in easier cases. Likewise, the tokenization experiments show that the models can tolerate segmentation changes, but they do not really reveal what internal computations make that possible. The discussion of RoPE as a possible explanation for the Shuffle Method also remains somewhat speculative, since the paper does not include a controlled comparison with models that use different positional encodings.

3. There is a meaningful confound in the cross family comparison. According to Appendix C, DeepSeek and Qwen are prompted in Chinese, while Llama is prompted in English. Even if the prompts are intended to be equivalent, prompt language itself could affect robustness under character level corruption. As a result, the conclusion that the Llama family is less robust than the other families is not fully disentangled from this factor.

4. The response classification used in the explicit rewriting analysis is not sufficiently validated. In Section 5.2.1 and Appendix C.3, responses from Qwen3 30B are categorized into A, B, and C using the same Qwen3 30B model. Since this analysis is one of the main pieces of evidence for the paper’s mechanistic claims, I would have liked to see some validation, such as human annotation or inter annotator agreement. Without that, the conclusions drawn from the classification feel less reliable.

---

> ### Author Rebuttal · Authors · 2026-03-31
>
> We sincerely thank you for your constructive feedback and for recognizing the originality of our framing, the strength of our empirical results, and the value of comparing multiple perturbation types within a unified setup. We have carefully addressed your concerns and added a new mechanistic experiment to strengthen our claims.
>
> **For weakness 1**，to address the concern that GSM8K’s structure might uniquely aid text recovery, we extended our evaluation to **[SQuAD](https://huggingface.co/datasets/rajpurkar/squad)**,  a reading comprehension dataset. The **results align with our initial findings**: the robustness persists, and performance metrics decrease as perturbation intensity increases. Models still demonstrate a strong ability to handle the Shuffle Method in this reading comprehension setting. The result confirms that the phenomenon generalizes beyond math problems. The results are shown in the table. Due to the limited time during the rebuttal period, we evaluated our methods on a randomly sampled subset of 1,000 examples from the dev set.
> | Metrics | Baseline | 30-Typo | 50-Typo_50 | 70-Typo | 100-Typo |
> |:-:|:-:|:-:|:-:|:-:|:-:|
> | em | 63.2 | 30.67 | 17.27 | 5.6 | 0.33 |
> | f1 | 81.48 | 51.18 | 33.31 | 13.25 | 1.39 |
>
>
> | Metrics | Baseline | 30-Shuffle | 50-Shuffle | 70-Shuffle | 100-Shuffle |
> |:-:|:-:|:-:|:-:|:-:|:-:|
> | em | 63.2 | 35.2 | 24.87 | 14.0 | 6.07 |
> | f1 | 81.48 | 56.26 | 42.84 | 27.16 | 11.18 |
>
>
> | Metrics | Baseline | 1-INJ | 2-INJ | 3-INJ |
> |:-:|:-:|:-:|:-:|:-:|
> | em | 63.2 | 7.67 | 0.73 | 0.2 |
> | f1 | 81.48 | 9.23 | 1.02 | 0.39 |
>
> **For weakness 2**, we agree that our discussion attributing Shuffle Method performance to RoPE lacked empirical support. Since it is currently infeasible to find a non-RoPE model with comparable capabilities to establish a controlled baseline, **we will remove the speculative discussion** from the paper. At mechanism level, we conduct a new experiment to export the internal mechanism on denoing, as descriped below.
>
> **For weakness 3**,  we clarify that **instruction prompts were not perturbed** during our tests; noise was only applied to the questions. We used English prompts for Llama due to its stronger native instruction-following capabilities in English. To ensure language was not a confounder, we tested Llama with Chinese prompts on a GSM8K subset. The performance did not significantly improve. This confirms that Llama's relatively lower robustness is intrinsic, rather than a prompt language artifact.
>
> **For weakness 4**, we agree on the importance of validating the automated classification. We **manually inspected** a subset of the Qwen3-30B self-annotations and confirmed that the model faithfully categorized the explicit rewriting behaviors. Furthermore, we employed methods to  enhance and stabilize the annotation accuracy, such as 1-shot example.
>
> **Furthermore, at the mechanistic level, we have supplemented our study with an additional experiment**:
> Using a noun with clear meaning(e.g., "river"), we generated two sentence sets—one ending with the target word, the other with a related word (e.g., "bay", "lake"). We split data 8:2 for training/testing. For each layer of Qwen3-8B, we trained a linear probe (logistic regression) to classify whether the hidden state encodes the target word. Perturbations were applied to the test set, and probes were used during forward passes.
> We hypothesized that the model internally denoises across layers: low accuracy in shallow layers, then a rise in middle-to-late layers as information is restored. Results support this:
> - **Baseline** (unperturbed): ~90% accuracy across all layers.
> - **100-Shuffle**: 50%in shallow layers, then rises significantly to ~80% in middle-late layers, dropping below 70% in final layers.
> - **50-Typo**: similar trend but less pronounced.
> - **Character-level tokenization**: accuracy rises at layer 3 and stays high.
> - **1-Inj**: similar to character-level, but recovery starts later (layer 6).
>
> For different cases, the accuracy curves of the classifier show some variation; nevertheless, they consistently adhere to the pattern of low initial accuracy, a subsequent increase in the middle-to-late layers, and a final decline. This experiment reveals that an internal mechanism capable of autonomously completing the denoising process indeed exists within the model.
>
> Here is the visual plot of our case study "river": ![](https://ibb.co/h1xT3Qkm)

---

> > ### Author Rebuttal · Reviewer_txk4 · 2026-04-04
> >
> > I thank the authors for their rebuttal. The authors have addressed most of my concerns.

---

> > > ### Author Response · Authors · 2026-04-05
> > >
> > > We sincerely thank you for your valuable comments and positive feedback. In light of your "Fully resolved" acknowledgment, would you kindly consider raising your score accordingly? We remain open to any further questions or suggestions you may have. Thank you again for helping us to improve our work!

---

### Decision · Program_Chairs · 2026-04-30

**Decision:**

Accept (regular)

**Comment:**

The recommendation is based on the reviewers' comments, the area chair's evaluation, and the author-reviewer discussion. This paper studies the effects of a set of LLMs under character-level perturbations. All reviewers find the studied setting novel and the results provide new insights. The authors’ rebuttal has successfully addressed the major concerns of reviewers.

In the post-rebuttal phase, most reviewers were satisfied with the authors’ responses and agreed on the decision of acceptance.
However, many reviewers share a common concern about
- Limited evaluation: the results may not generalize to other untested LLMs
- Confounded effect: use of different prompt languages (this was addressed in the rebuttal)
- Lack of mechanistic understanding of the observed effects

That being said, I also want to make a note that
- One reviewer mentioned increasing the score from 4 to 5, yet the review has not been updated
- One reviewer who originally gave Reject found the rebuttal fully addressed the concerns, yet did not update the review rating

Overall, this is a borderline paper (not top 30%) in my batch. I recommend weak acceptance of this submission. But I wouldn't mind if the paper got rejected if SAC deems it necessary.